# Location is a major barrier for transferring US fossil fuel employment to green jobs

Junghyun Lim[1], Michaël Aklin[2,3] & Morgan R. Frank [4,5,6] ✉

The green energy revolution may displace 1.7 million fossil fuel workers in the US but a Just Transition to emerging green industry jobs offers possibilities for re-employing these workers. Here, using 14 years of power plant data from the US Energy Information Administration, job transition data from the Census Bureau, as well as employment and skills data from the Bureau of Labor Statistics, we assess whether people employed in fossil fuel resource extraction today are co-located and have the transferable skills to switch to expected green jobs. We find that these workers could leverage their mobility to other industries and have similar skills to green occupations. However, today's fossil fuel extraction workers are not co-located with current sources of green energy production. Further, after accounting for federal employment projections, fossil fuel extraction workers are mostly not located in the regions where green employment will grow despite attaining the appropriate skillsets. These results suggest a large barrier to a Just Transition since fossil fuel extraction workers have not historically exhibited geospatial mobility. While stakeholders focus on re-skilling fossil fuel extraction workers, this analysis shows that co-location with emerging green employment will be the larger barrier to a Just Transition.

Climate change is contributing to more storms, flooding, and wildfires thus underscoring the need for policy to combat rising temperatures. Without effective carbon dioxide removal technology, limiting global warming to 2 degrees Celsius requires a phaseout of fossil fuels by 2050[1,2]. However, such a phase would incur several social and economic consequences beyond its climatic implications. For example, this transition could displace 1.7 million fossil fuel workers in the United States[3] and many more globally[4]. A phaseout could also have socio-economic consequences for local communities whose economies are strongly dependent on local fossil fuel employment[5,6] including employment loss, suppressed economies, and negative impacts to community ideology. Past phaseouts have proven challenging[7,8] as large-scale labor market transitions are notoriously costly and difficult to complete. Accordingly, workers, unions, and policymakers are seeking a "Just Transition" in which fossil fuel

workers receive public support to find new jobs[9–14]. The Just Transition is an umbrella concept that connects both normative concerns about the ethics of existing and alternative energy systems as well as positive debates about the governance underpinning the clean energy transition[15]. Among these, the challenge faced by fossil fuel workers has received increasing attention as one of the constituencies that are directly affected by policy decisions[16–20].

Stakeholders have flagged green jobs as a potential solution. For instance, the Biden administration's *Inflation Reduction Act* could create 9 million jobs[21] connected to its climate, energy, and environmental justice programs. This strategy could absorb displaced fossil fuel workers[22]. Likewise, the European Union has a €55b (about $52b) Just Transition Mechanism, part of which is devoted to helping fossil fuel regions build out new industries. Similar calls for public support of a fossil-to-green-job pipeline have been made in

[1]Department of Political Science, University of North Carolina at Chapel Hill, Chapel Hill, NC 27514, USA. [2]PASU Chair, College of Management of Technology, EPFL, 1015 Lausanne, Switzerland. [3]Enterprise for Society, 1015 Lausanne, Switzerland. [4]Department of Informatics and Networked Systems, University of Pittsburgh, Pittsburgh, PA 15216, USA. [5]Digital Economy Lab, Institute for Human-Centered Artificial Intelligence, Stanford University, Stanford, CA 94305, USA. [6]Media Laboratory, Massachusetts Institute of Technology, Cambridge, MA 02139, USA. ✉e-mail: mrfrank@pitt.edu

China, India, South Africa, and other large greenhouse gas emitters[23–26].

How realistic is a reassignment of fossil fuel workers to green jobs? Stakeholders, including the Biden administration, assert that fossil fuel workers are appropriately skilled for emerging green jobs[27–29]. But displaced workers must also co-locate with new jobs for transitions to occur[30–33]. It is not definitive that either requirement is met. For example, Western Pennsylvania is specialized in excavation workers for mining (as shown by its location quotient: $LQ = 11.13$) as well as service unit operators ($LQ = 7.04$) and rustabouts ($LQ = 4.87$) in oil and gas according to the US Bureau of Labor Statistics (BLS). But it is an unlikely region for solar technology jobs because of cloud coverage[18]; Pittsburgh, PA, is routinely among the top ten cloudiest US cities according to the National Oceanic and Atmospheric Administration. The problem of geographical co-location between fossil and green jobs has started to receive more attention, but existing studies typically do not examine the broader portfolio of occupations, nor do they estimate the mobility of fossil fuel workers[18,19]. And, even if green jobs emerge, how will skill similarity between fossil fuel and green industry occupations shape job transitions[7,34,35] and the resilience of labor markets[36,37]?

Although workers' individual job prospects and experiences will vary, this study considers a macro-economic perspective on career opportunities for fossil fuel workers in emerging green jobs. We approach the problem using occupation skill profiles from the US Bureau of Labor Statistics (BLS), data on current solar, hydro, wind, and biomass power plants from the US Energy Information Administration, and BLS employment projections to estimate green job growth. First, we compare the skill requirements of fossil fuel workers to BLS-identified green occupations and find that these occupations indeed require similar skills which suggests that a Just Transition may occur without large re-skilling efforts. Second, we examine if today's fossil fuel workers are co-located with current green energy-producing power plants and find little co-location. This echoes earlier studies using finer-grained data in terms of geographical and occupational information[18]. Third, since green jobs may emerge in new places over the next decade, we employ BLS employment projections to estimate green job growth across US urban and rural regions. But, again, we find little co-location between 2019 fossil fuel worker employment and projected green job growth. Fourth, we examine worker mobility data

by industry provided by the US Census Bureau to find that relocation of fossil fuel workers have been limited historically. Combined, these results highlight that relocation may be a barrier to a Just Transition for US fossil fuel workers despite having the appropriate workplace skills. Finally, we examine how targeted investments in fossil fuel dominated labor markets might mitigate the need for workers to relocate. Our findings demonstrate that today's fossil fuel workers are, mostly, appropriately skilled for green occupations but are not located in the regions where green jobs are likely to emerge. Thus, prudent policy in support of a Just Transition must address this co-location issue either by creating incentives for today's fossil fuel workers to relocate or by stimulating new employment opportunities in the regions where fossil fuel workers currently reside.

## Results

Recent studies[38–41] show that skill similarity mediates workers' transitions between jobs in general. Here, we use occupation skill profiles from the O*NET database used by BLS to compare green occupations[42] with fossil fuel industry occupations and other two-digit NAICS sectors (see SI sections 1 and 2). Among the fossil fuel industry's occupations, we focus on extraction workers, who represent the core and the largest group of occupations within that sector (see SI Section 1 for a list of occupations and analysis with alternative specifications). To identify green occupations, we rely on existing classifications[42] which have been widely used by the US Bureau of Labor Statistics (BLS) and the European Union[43]. Among the types of green occupations, we focus on occupations that are classified to be emerging in the renewable energy sector that contribute to the reduction of fossil fuel emissions. We analyze alternative green occupation specifications in SI Section 2. We describe employment and skills data in the Methods Section and SI Section 3.

Will fossil fuel workers need re-skilling to perform green jobs? We compare the skill requirements of fossil fuel occupations to the occupations in other industries using Jaccard similarity (denoted *skillsim*. See equation (3) in Methods). The *skillsim* function ranges from 0 to 1, with higher values denoting greater similarity between sets of required skills. Indeed, fossil fuel workers, $f$, have significantly more skill similarity to green industry occupations, $g$, than to other industries according to a two-sample $t$-test ($p < 0.0001$) with an average score of $skillsim(f, g) = 0.79$ (see Fig. 1A). However, according to

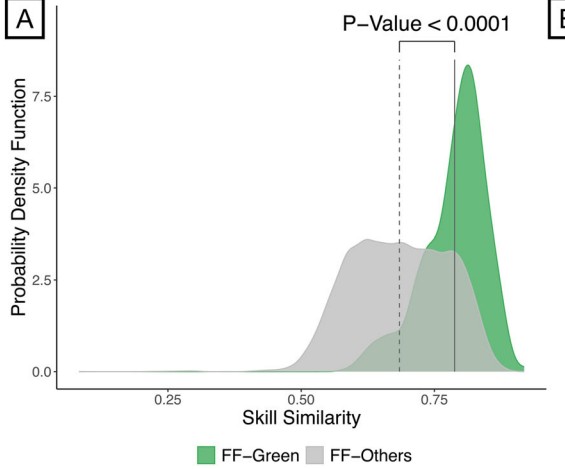

| | Model 1 | Model 2 | Model 3 | Model 4 | Model 5 |
|---|---|---|---|---|---|
| | \multicolumn{5}{c}{Dependent Variable: $Transition_{f,m,i',m'}$} | | | | |
| Skill Similarity$_{i,i'}$ | | 0.59 (0.0001) $P \leq 10^{-3}$ | | 0.84 (0.0001) $P \leq 10^{-3}$ | 0.41 (0.0003) $P \leq 10^{-3}$ |
| Distance$_{m,m'}$ | | | -1.13 (0.0001) $P \leq 10^{-3}$ | -1.18 (0.0001) $P \leq 10^{-3}$ | -2.07 (0.0003) $P \leq 10^{-3}$ |
| Employment$_{f,m}$ | 0.94 (0.0002) $P \leq 10^{-3}$ | 0.97 (0.0002) $P \leq 10^{-3}$ | 1.01 (0.0002) $P \leq 10^{-3}$ | 1.00 (0.0002) $P \leq 10^{-3}$ | 1.04 (0.0002) $P \leq 10^{-3}$ |
| Employment$_{i',m'}$ | 0.85 (0.0002) $P \leq 10^{-3}$ | 0.90 (0.0002) $P \leq 10^{-3}$ | 0.98 (0.0002) $P \leq 10^{-3}$ | 0.97 (0.0002) $P \leq 10^{-3}$ | 1.04 (0.0002) $P \leq 10^{-3}$ |
| Stay (Industry) | | | | | 1.11 (0.0006) $P \leq 10^{-3}$ |
| Stay (Location) | | | | | -3.43 (0.0012) $P \leq 10^{-3}$ |
| Constant | 1.16 (0.0002) $P \leq 10^{-3}$ | 0.95 (0.0003) $P \leq 10^{-3}$ | 0.23 (0.0003) $P \leq 10^{-3}$ | -0.04 (0.0003) $P \leq 10^{-3}$ | -0.34 (0.0003) $P \leq 10^{-3}$ |
| Pseudo $R^2$ | 0.16 | 0.21 | 0.72 | 0.81 | 0.84 |
| Observations | 10,352,319 | 10,352,319 | 10,352,319 | 10,352,319 | 10,352,319 |

**Fig. 1 | Although both are significant, geospatial distance is a bigger factor than skill similarity in fossil fuel worker mobility. A** Fossil fuel workers' skills are more similar to the skill requirements of green jobs than to those of other industries according to a two-sample $t$-test. $p$-value is <0.001 and a 95% confidence interval of difference in mean is [0.099, 0.108]. **B** We use a Poisson model to predict the flows of workers who transition from the fossil fuel industry ($f$) in metropolitan area $m$ to industry $i$ in metropolitan area $m'$ according to industry-region migration data from

the US Census Bureau from 2005 to 2019. Distance and employment are log-transformed. The Stay (Industry) and Stay (Location) indicator variables control for workers who remain in the same industry or MSA/NMSA, respectively. All variables are centered and standardized (i.e., transformed into z-scores) so regression coefficients are directly comparable. Coefficients are reported, followed by standard errors in parentheses and $p$-values.

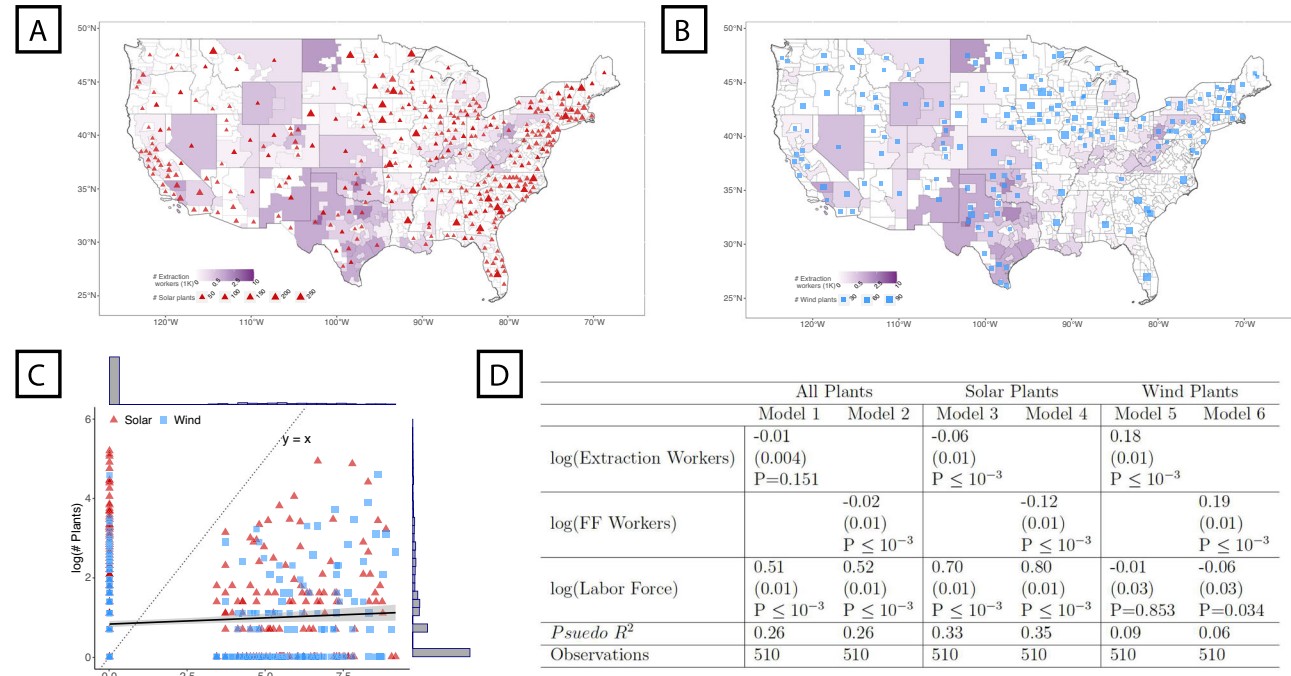

**Fig. 2 | Green energy plants are not co-located with today's fossil fuel workers.** We map the location of the current extraction workers (purple), (**A**) solar energy plants, and (**B**) wind energy plants. **C** A scatter plot comparing the number of extraction workers (x-axis) to the number of power plants (y-axis) by power plant type (color). Most regions contain either green energy power plants or extraction workers, but not both (see histograms). We estimate the line of best fit along with a 95% confidence interval (see solid line). **D** Poisson regressions investigating the 2019 correlation between fossil fuel worker employment and the number of green energy plants while controlling for the size of the local labor market. Coefficients are reported, followed by standard errors in parentheses and *p*-values. We provide similar analyses for hydro and biomass power plants in Section 5 of the Supplementary Materials. Maps were made using the sf package in R (Pebesma E (2018). "Simple Features for R: Standardized Support for Spatial Vector Data." The R Journal, 10(1), 439-446.--CC-BY Attribution 4.0).

industry-region migration Job-to-Job (J2JOD) data from the US Census Bureau spanning 2006 to 2019, fossil fuel workers become more likely to transition to industries with $skillsim(f, i) \geq 0.9$ (see Fig. 10 in SI section 4). Thus, some re-skilling may be required even though fossil fuel workers' skills are better matched to green occupations than to other industries.

Overall, how critical is skill similarity to fossil fuel worker mobility? We analyze the historical flow of fossil fuel workers to other industries and regions (i.e., metropolitan statistical areas, denoted MSAs, and non-metropolitan statistical areas, denoted NMSAs) with a Poisson regression of the J2JOD data from the Census (see Fig. 1B; see Methods for more information on J2JOD data). All variables are centered and standardized (i.e., transformed to z-scores) to remove units so that coefficient estimates are comparable across variables (i.e., coefficients represent changes in standard deviations). This normalization enables us to identify how a standard deviation change in a variable (e.g., skill similarity) corresponds to standard deviation changes in the worker flows from fossil fuel occupations to other industries, and, in particular, it allows us to compare which variables are most strongly associated with fossil fuel worker mobility relative to each variable's natural variability. As a baseline, we first consider a random mixing model based on regional employment by industry (see Model 1). Adding skill similarity to the baseline model yields a 31% factor improvement in predictive performance (see Model 2). However, adding distance to employment (i.e., the gravity mobility model) achieves a pseudo-$R^2$ of 0.72 and distance is the most important factor in the model (see Model 3). Adding skill similarity to the gravity model captures more information about fossil fuel worker mobility and increases the pseudo-$R^2$ to 0.81 (i.e., an additional 9 percentage points of variation explained; see Model 4). However, distance continues to be the most important factor in the model. Compared to the baseline random mixing model (Model 1), including geospatial distance (Model

3) improves pseudo-$R^2$ by a bigger margin than adding skill similarity to the model (Model 2). These results hold after controlling for fossil fuel workers who do not change industries or do not relocate (see Model 5).

Since distance has been a barrier to fossil fuel workers' mobility, are today's fossil fuels workers co-located with green jobs? Investigating this question is challenging. While green employment is expected to grow over the next decade, it is hard to forecast where new jobs will appear. Therefore, we first consider the locations of today's green energy producing power plants using data from the US Energy Information Administration to approximate the labor markets currently supporting green jobs. In Fig. 2 A&B, we map the locations of today's solar and wind power plants and compare them to the 2019 employment distribution (i.e., before the COVID pandemic) of fossil fuel workers in MSAs and NMSAs (see SI section 5 for a similar analysis of other energy sources). In both cases, this correlational analysis reveals that the number of plants does not strongly correlate with fossil fuel worker employment according to a cross-sectional comparison. Most regions are dominated by only power plants or fossil fuel worker employment, but not both (see Fig. 2C). A Poisson regression that accounts for labor market size reveals only weak associations between the spatial distribution of green energy power plants and fossil fuel workers (see Fig. 2D).

Analyzing current power plant locations provides a static view of the co-location barrier to a Just Transition for fossil fuel workers, but future green jobs may emerge in new regions as a result of federal government funding to support green industry growth (e.g., the Inflation Reduction Act). Accordingly, we employ 2029 employment projections from the US BLS combined with historical employment data to predict where new green jobs might emerge. We train a random forest regression[44–46] to predict green employment in each MSA and NMSA in 2029 (i.e., 10 years after the last non-COVID year; see SI

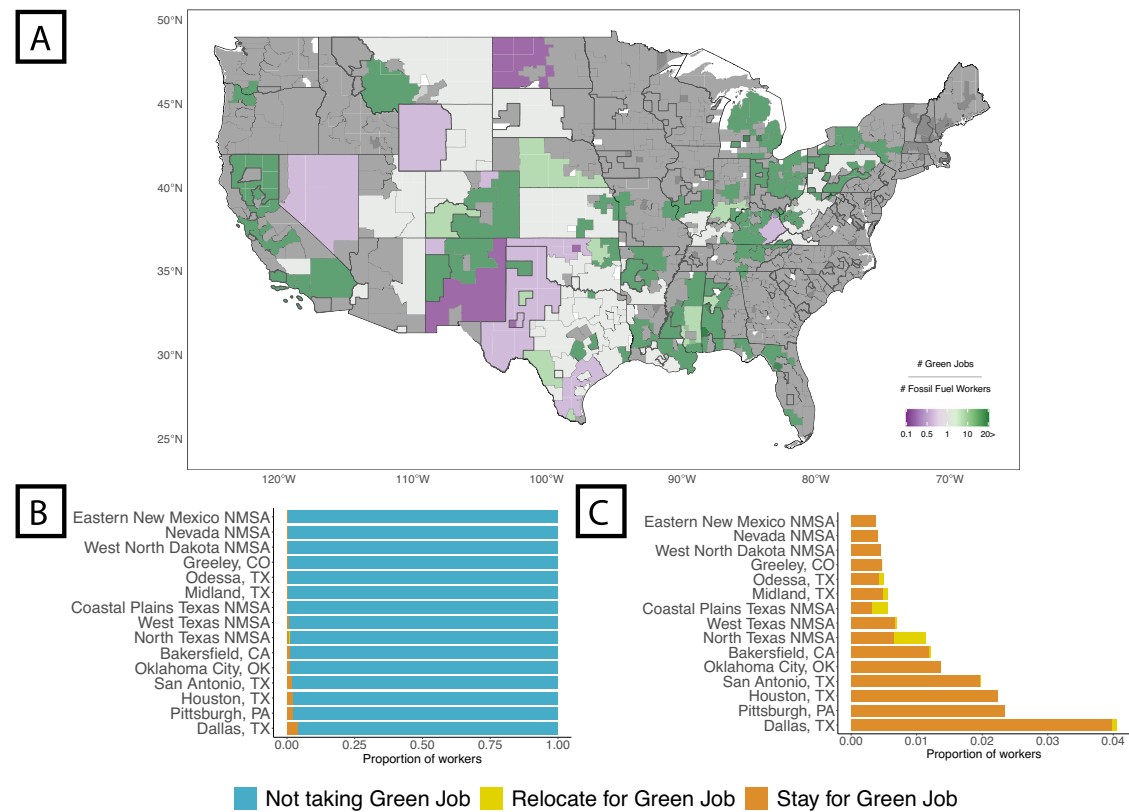

**Fig. 3 | US fossil fuel workers are not co-located with projected green job growth. A** The ratio between expected green jobs and 2019 fossil fuel worker employment. Maps were made using the sf package in R (Pebesma E (2018). "Simple Features for R: Standardized Support for Spatial Vector Data." The R Journal, 10(1), 439–446. ·CC-BY Attribution 4.0). **B, C** In the 15 most extraction-intensive regions, we expect that <1.5% of fossil fuel workers will transition to green jobs. Even in regions with the highest transition rates (i.e., in Dallas, TX), only 4% of fossil fuel workers will transition to green jobs.

section 2). We validate our approach using historical data and achieve out-of-sample performance of $R^2 = 0.82$ and out-of-sample RMSE = 0.57 following cross-validation. We then apply our model to 2019 data and 2029 BLS employment projections to estimate the green job growth across US regions in 2029. We predict the transition of fossil fuel workers to green jobs by combining predicted employment growth in each region with Model 5 from Fig. 1B. Several regions within Great Plains states will have green employment that is comparable their local fossil fuel employment in 2019 (see Fig. 3A). However, many of the regions with the greatest number of fossil fuel workers, including regions in Nevada, New Mexico, Western Pennsylvania, and North Dakota, will not experience comparable green job growth.

In total, the vast majority of extraction workers (98.97%) will not transition to green jobs according to our model. In an idealized scenario where all the green jobs are co-located with fossil fuel jobs, our model predicts that 13.7% of extraction workers will transition. In another idealized scenario where fossil fuel workers match green occupations' skills exactly (i.e., *skillsim* = 1) and all else being equal, our model predicts that 5.51% of extraction workers will transition to green jobs. Thus, while both skill similarity and spatial distance play important roles, geospatial distance is the primary barrier to transitions. These findings are consistent if we focus on the 15 regions with the most extraction workers and largest quantities of fossil fuel production (see Fig. 3B, C). Among fossil fuel workers who transition to green jobs, a majority will do so without relocating. The estimated job transitions from fossil fuel to green jobs for these regions show the impact of geographical constraints with few transitions expected beyond 20 miles from a worker's point of origin (see Fig. 4A). This observation is reinforced when we look at the likely transitions in three labor markets: Bakersfield CA, North Texas NMSA, and Pittsburgh PA (see Fig. 4B).

Most workers are expected to stay within their current labor market. This limited mobility of workers suggests the importance of the location of future green employment.

We simulate several scenarios where new green jobs are created and distributed either in proportion to total employment in every region or proportionally to 2019 fossil fuel employment. Figure 11 in the Supplementary Information highlights the differences in spatial distributions of green jobs under these different scenarios. Applying the same transition model (see Fig. 1B, Model 5), we find that the share of fossil fuel workers who are expected to transition to green jobs are higher in scenarios where creation of green jobs are geographically targeted to regions in proportion to existing fossil fuel employment (i.e., geo-targeted scenarios in Fig. 4C). For example, creating 1 million new green jobs across regions in proportion to existing fossil fuel employment would result in higher transition rates for fossil fuel workers to green jobs compared to a scenario where 5 million new jobs are distributed across regions in proportion to their total employment (i.e., compare geo-targeted (1M) and non-targeted (5M) scenarios in Fig. 4C).

Our analysis is robust to varying definitions for green industry occupations, but other existing industries may also absorb displaced fossil fuel workers and see employment growth that complements a growing green industry. For example, green industry growth may lead to employment growth in Manufacturing as demand for new green energy power stations increases. Thus, in addition to green jobs, we explore several scenarios where fossil fuel extraction workers transition to jobs in non-green industries. We consider the three target industries with the greatest skill similarity to fossil fuel occupations (i.e., Construction, manufacturing, transportation and warehousing) and the three target industries with the least skill similarity to fossil fuel

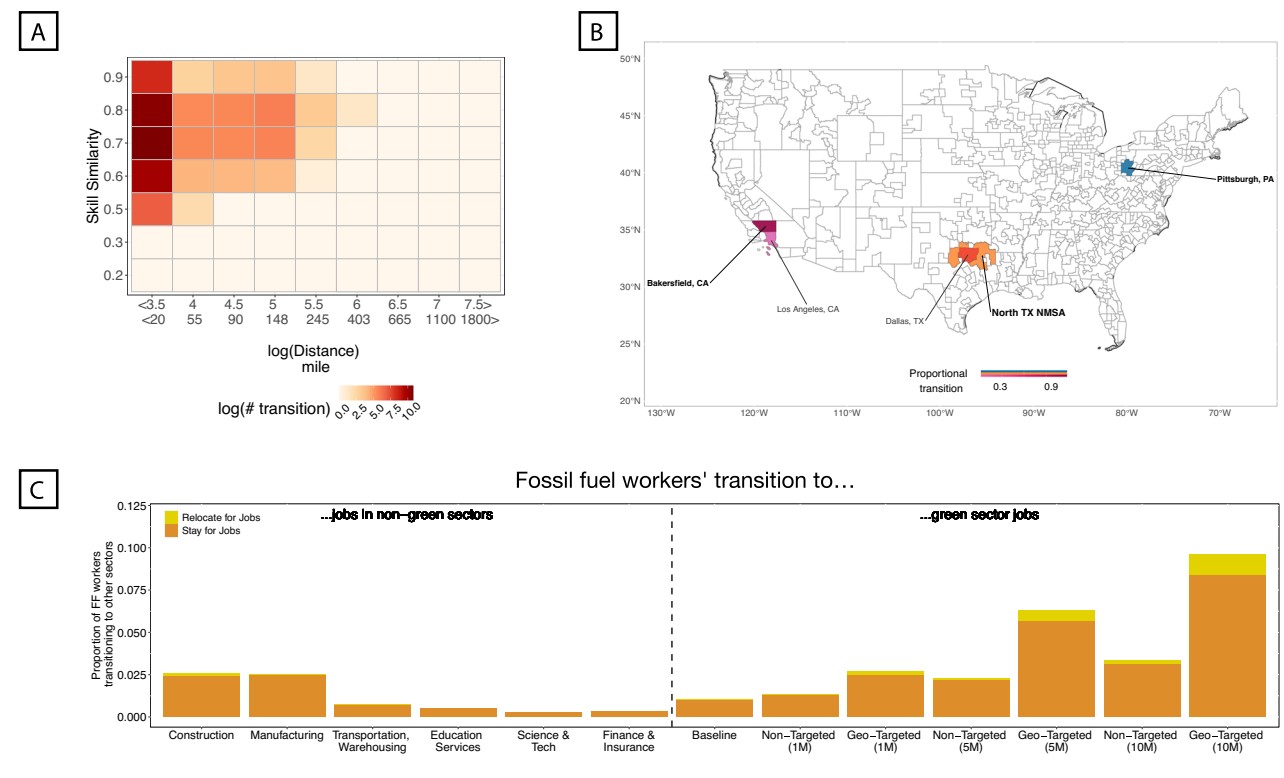

**Fig. 4 | Among fossil fuel workers expected to transition to green jobs, only a small share will relocate. A** A heat map detailing skill similarity and geospatial distance in expected fossil fuel worker transitions to green jobs. Transitions are concentrated around small distances towards the left of the plot. **B** Three examples of the spatial dispersion of fossil fuel workers from extraction to green jobs. The spatial range of dispersion is small in each case. Maps were made using the sf package in R (Pebesma E (2018). "Simple Features for R: Standardized Support for Spatial Vector Data." The R Journal, 10(1), 439-446. -CC-BY Attribution 4.0). **C** The predicted proportion of fossil worker transitions to different existing sectors and green jobs. For existing sectors, we consider three sectors with the highest and lowest skill similarities with fossil-fuel extraction workers respectively (SI Section 3). For green jobs, in addition to the baseline prediction, we consider potential policy interventions to promote green job growth (e.g., the Biden Administration's Inflation Reduction Act). We explore scenarios of 1, 5, and 10 million new green jobs distributed either proportionally to 2019 fossil fuel worker employment (Non-Targeted) or proportional to regions' total employment (Geo-Targeted).

occupations (see SI section 4.1). Using our job transition model (i.e., Fig. 1B, Model 5), we find that transition rates of fossil fuel workers are higher for manufacturing and construction sectors compared to transition rates to green occupations assuming no policy intervention (baseline scenario in Fig. 4C). Even when considering a scenario where 5 million new jobs distributed in proportion to total employment (non-targeted (5M) in Fig. 4C), transition rates to manufacturing and construction sectors are higher than transitions to green jobs. However, when we consider one million new green jobs created in proportion to existing fossil fuel employment (geo-targeted (1M) in Fig. 4C), we find similar transition rates to manufacturing and construction, and with 5 million new green jobs in fossil fuel-intensive regions, the transition rates are even higher than in both construction and manufacturing scenarios. These results emphasize the importance of co-locating new employment opportunities with existing fossil fuel workers and, further, demonstrate that other existing industries may be viable options for absorbing displaced fossil fuel workers.

Our study employs forecasts of green job growth (see SI section 2 for cross-validation with historical data), but we find similar results using several alternative estimates. For example, aggregating our sub-state regional estimates of green job growth by state reveals a strong agreement with green employment estimates from the Princeton Net-Zero America Project[47] (see SI section 2.2.2).

## Discussion

Can a Just Transition be achieved by transitioning fossil fuel workers to new green jobs? The transferability of fossil fuel workers' skills is one factor in their possible transition to other industries. Largely, stakeholders assume that fossil fuel workers have the skills for green jobs while ignoring where green job growth might occur. Our results highlight that fossil fuel workers have greater skill similarity to green industry occupations than to other industries, but further re-skilling might improve the transition. However, co-location between today's fossil fuel workers and emerging green jobs will be the larger barrier to a Just Transition. Historically, fossil fuel workers have not exhibited the geospatial mobility necessary to absorb today's fossil fuel workers into the green industry.

Low spatial mobility among fossil fuel workers has important implications for policy design. For example, in line with earlier studies[14], we simulate how a policy creating green jobs (e.g., the Inflation Reduction Act in 2022) could impact fossil fuel workers depending on whether it targets markets with fossil fuel workers or not (see Fig. 4C). For a comparable number of new jobs, geo-targeted interventions create more mobility to green jobs thus underscoring the low mobility among fossil fuel workers. In particular, a targeted intervention that creates one million jobs will be more helpful to fossil fuel workers than a non-targeted policy that creates five million positions despite creating more new jobs. This reinforces the importance of the location of green job growth.

Our modeling approach makes several assumptions that limit our analysis. First, we have assumed that all of today's fossil fuel workers will want to transition to green jobs. However, there may be additional social barriers including workers' preferences[48], identity, culture[49], and economic outlook[50,51]. Second, we have assumed that fossil fuel

workers share the same level of latent mobility. We train a model to predict the average mobility of fossil fuel workers using industry level mobility data from the US Census Bureau (see Fig. 1B for analysis of J2JOD data) and then apply this model to the occupation level data (skill similarity, employment) to estimate the proportion of workers transitioning to green jobs. However, average behavior may obfuscate other mobility dynamics among fossil fuel workers, particularly across sub-sectors, influenced by socio-economic or cultural factors. Third, workers from other industries may compete with fossil fuel workers to fill green industry jobs. Fourth, we have looked at a narrow subset of extraction workers but other workers in the fossil fuel sector, such as engineers and lawyers, would also need to find new jobs if the sector phased out entirely. Fifth, currently unknown occupations may emerge as the green industry continues to evolve. Finally, other barriers—particularly social barriers beyond employment—may shape a Just Transition but are not addressed in this study. With these limitations in mind, our results represent a best-case scenario for fossil fuel workers even though there exist additional barriers to a Just Transition. Because of these necessary assumptions, we include several alternative approaches to our analysis as robustness checks. For example, we vary our definition of "green occupation" (see SI section 2.1), our estimates of future green employment growth vary from a conservative approach based on the location of current green energy power plants to a more complex approach based on employment predictions from the US Bureau of Labor Statistics. Further, we consider the potential impact of federal stimulus in fossil fuel communities (see Fig. 4C).

One way to achieve a Just Transition is to generate a sufficient number of green industry jobs. But the issue is more complicated than simply the total number of job opportunities that will be created. To be successful, this transition requires a high degree of skill similarity and geographical congruence between green and fossil fuel jobs—in addition to solutions for other non-economic social barriers. We find considerable evidence that skills in both sectors match well. But our study generates more concerns about geographical frictions. From a policy standpoint, addressing geographic mismatch ought to become a priority. Practically, this can be done in two ways: facilitate relocation of fossil fuel workers or attract new industries to the regions they live in already. These observations echo debates between place- and people-based investments and the optimal design of industrial policy[52–54]. Future work may thus pay close attention to two problems. The first is to understand the social factors that shape workers' hesitancy to relocate. The second is to identify the optimal diversification strategies *given* the existing distribution of workers[55]. This has informed recent policy and regulatory initiatives. For example, the Inflation Reduction Act recognizes the benefits of incentivizing investments in so-called "energy communities." Yet further refinements may be necessary to optimize the chances of a successful transition. Otherwise, multi-billion dollar policy initiatives may become costly wasted opportunities.

## Methods
### Occupation and skills data
We use occupation skill profiles from the O*NET database. O*NET is an annually updated database produced and maintained by the US Bureau of Labor Statistics (BLS). The BLS produces these skill profiles using surveys of actual workers and assessments from BLS analysts. O*NET provides information on required workplace skills for over 750 occupations defined by Standard Occupation Classification (SOC) taxonomy. The BLS uses SOC codes to also report employment distributions for regions (e.g., MSAs, NMSAs, states, and nationally). O*NET identifies the 232 different skills and rates their importance and the level of required skill in each occupation either in continuous range of 1–5 or 0–7. We normalized these metrics to a continuous range of 0 to 1, where 1 is the most important skill for a given occupation. O*NET

data is widely used by the US Department of Labor, policy makers, and researchers to study skills in the US workforce. For example, an economics study from the OECD used O*NET data to estimate automation from technology[56]. As another example, O*NET data is used in a US Department of Labor report to identify green occupations[42]. SI Tables 3, 4, 5, and 6 detail the most important skills for fossil fuel occupations and green industry occupations.

### Measuring skill similarity
We use $onet_{j,s} \in [0,1]$ to denote the importance of skill $s \in S$ to occupation $j \in J$ such that $onet_{j,s} = 1$ identifies an essential skill and $onet_{j,s} = 0$ indicates an irrelevant skill. Using this data, we calculate the skill similarity for a pair of occupations $j$ and $j'$ according to the real-valued Jaccard similarity:

$$skillsim(j,j') = \frac{\sum_{s \in Skills} \min(onet_{j,s}, onet_{j',s})}{\sum_{s \in S} \max(onet_{j,s}, onet_{j',s})}. \quad (1)$$

Note that $skillsim(j,j') = 1$ when occupations $j$ and $j'$ require the exactly same skills while $skillsim(j,j') = 0$ when occupations $j$ and $j'$ require completely different skills.

### Job transition data
The Job-to-Job flow origin-destination (J2JOD) data by the US Census Bureau provides the quarterly flow (count) of workers who transition from one job to another. The data includes information on the original sector and the worker's location at the metropolitan statistical area level as well as the location and sector of a person's new job. Sectors are defined following the North American Industry Classification System (NAICS) 2 digit code.

### Employment data
The BLS provides detailed data on the size of the workforce in each occupation defined by SOC taxonomy (6-digit) at metro and non-metropolitan areas from 2005 to recent years. Using this data, we identify the number of workers who work in extraction occupations, and potential green occupations defined by[42]. Additionally, we use the distribution of the entire workforce fossil-fuel industry using data from the Business Dynamics Survey (BDS) from Census Bureau. BDS provides the number of workers in metropolitan areas at the industry level, following the NAICS 2 digit code. Finally, BLS provides employment breakdown by occupation (SOC 6 digit) and industry (NAICS) at the national level. Using this employment breakdown data combined with the occupational skill similarity measure, we compute the industry level skill similarity scores (See Supplemental Material Section 3.3).

### Comparing skill requirements across industries
Whereas we primarily use the occupation-level skill similarity to produce predictions for workers' mobility, we also employ the skill similarity across industries to train a model to predict the transition of workers across industries using J2JOD (Job-to-Job Flow, Origin-Destination) data provided by the US Census Bureau. To estimate skill similarity at the sector level, we use the weighted average of the importance of each skill ($onet(j,s)$) with the share of employment in occupation $j$ in industry $i$. In other words, we measured the skill similarity between average workers in the fossil-fuel sector and other industry $i'$. To accomplish this, we represent the importance of skill $s$ to industry $i$ according to

$$onet(i,s) = \frac{\sum_{j \in Jobs} onet_{j,s} \cdot N_{j,i}}{\sum_{j' \in Jobs} N_{j',i}} \quad (2)$$

where $N_{j,i}$ is the employment count for occupation $j$ within industry $i$. Then, we calculate the skill similarity between industries $i$ and $i'$

according to

$$\text{Skill similarity}_{i,i'} = \frac{\sum_{s \in S} \min\left(onet(i,s), onet(i',s)\right)}{\sum_{s \in S} \max\left(onet(i,s), onet(i',s)\right)}. \quad (3)$$

Section 3.3 in the Supplementary Materials provides the distribution of industry-pair skill similarity scores and a list of the US sectors with the greatest skill similarity to fossil fuel occupations. Construction (NAICS 23) and Utilities (NAICS 22) have the greatest skill similarity to fossil fuel occupations.

## Modeling job transitions between industries
We estimate variations of the following model with 10-fold cross-validation:

$$\text{Transition}_{f,m,i',m'} \sim \lambda_{f,m,i',m'}$$
$$\log(\lambda_{f,m,i',m'}) = \beta_0 + \beta_1 \log(\text{Distance})_{m,m'} + \beta_2 \text{Skill similarity}_{f,i'} \quad (4)$$
$$+ \beta_3 \text{Employment}_{f,m} + \beta_4 \text{Employment}_{i',m'}$$

The distance between regions $m$ and $m'$ (i.e., MSAs or NMSAs) is calculated in miles from the centroid of the regions' Tiger Shape File provided by the US Census Bureau. Distances are calculated using the Haversine formula. The dependent variable in this analysis comes from the J2JOD data provided by the US Census Bureau and the employment statistics come from the US Bureau of Labor Statistics.

## Predicting green employment
We predict the distribution of green employment in 2029 using historical employment data from the BLS in metropolitan and non-metropolitan areas from 2005 to 2019, as well as other demographic data from Census (population by age, gender, race, education in each area), and economic features (state-level GDP per capita by industry defined by NAICS 2-digit). We train the model to predict the employment using the data from 10 years lagged. We compare the performance of random forest against other regressors (OLS, Lasso). Cross-validation (10 fold) was used to tune the hyper-parameters of the model. We use the historical employment data to train and test the accuracy of predictions. To avoid data leakage, we separated the training set and test set using the temporal cutoff. We set aside the 2019 employment distribution data as test set while training the model with the previous years.

## Projecting transition to green occupations
We predict the fraction of the extraction workers who will take green jobs using the model 5 in Fig. 1B, combining with the existing data on the number of the extraction workers, predicted value of green occupation in 2029, skill similarity between each occupation, and geographical distance as input. Here, we use the skill similarity measured based on 2019 O*NET data, under the assumption that skill requirements for each occupation in 2019 will remain the same in 2029.

## Simulating different green transition scenarios
In Fig. 4C, we simulate six different scenarios of future supply of green jobs by combining (1) varying size of additional green jobs with (2) whether the intervention is geographically targeted towards regions with a high number of fossil fuel workers or not. For non-targeted scenarios, we increase the number of fossil fuel jobs at the same rate for every area, based on our baseline predicted value of green jobs in 2029. In geo-targeted scenarios, we distribute the green jobs proportionally to the share of fossil fuel workers across metropolitan and non-metropolitan areas. The increase in the number of each green job ($i$) in each region ($m$) for different scenarios can be represented as

follows:

Non-Targeted:
$$GOCC_{i,m} + (\text{\# of Jobs created}) \times \frac{GOCC_{i,m}}{\sum_{i,m} GOCC_{i,m}} \quad (5)$$

Geo-Targeted:
$$GOCC_{i,m} + (\text{\# of Jobs created}) \times \frac{GOCC_{i,m}}{\sum_i GOCC_{i,m}}$$
$$\times \frac{\sum_i FOCC_{i,m}}{\sum_{i,m} FOCC_{i,m}} \quad (6)$$

where $GOCC_{i,m}$ as the predicted number of green job $i$ in area $m$ in baseline scenario, $FOCC_{i,m}$ as the number of fossil-fuel extraction job $i$ in area $m$ in 2019. In our analyses, we vary "# of Jobs created" by 1M, 5M, and 10M. Across different scenarios, we do not take into account specific occupation based targeting. In SI section 4.1, Figure 11 shows the different spatial distribution of future green jobs by whether policy interventions to create green jobs are geo-targeted (Fig. 11B) or not (Fig. 11A).

## Reporting summary
Further information on research design is available in the Nature Portfolio Reporting Summary linked to this article.

## Data availability
All data in this study is publicly available. The power plant location data is available through the US Energy Information Administration (https://eia.maps.arcgis.com/home/item.html?id=bf5c5110b1b944d299bb683cdbd02d2a). The J2JOD worker migration data is available through the US Census Bureau (https://lehd.ces.census.gov/data/). Employment data and occupation profiles are available through the US Bureau of Labor Statistics (https://www.bls.gov/). Replication data and code is available through figshare (https://doi.org/10.6084/m9.figshare.23907732). Authors will make additional data available upon request.

## Code availability
All code was produced in Python version 3 and R version 4.3. Replication data and code is available through figshare (https://doi.org/10.6084/m9.figshare.23907732).

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

## Acknowledgements

This work is funded by a grant from the Heinz Endowments (#E9694). The authors thank the participants at the 2023 Petralia workshop in

applied economics, Shanti Gamper-Rabindran, Matto Mildenberger, Esteban Moro, and Daniel Rock for their helpful feedback on this study.

## Author contributions

J.L., M.A., and M.R.F. designed the research and drafted the manuscript. J.L. performed all data analysis and generated all figures and tables. M.A. and M.R.F. secured funding in support for this project.

## Competing interests

The authors declare no competing interests.
