## [Peer Review File · Nature Communications]

Location is a major barrier for transferring US fossil fuel employment to green jobsEditorial Note: Parts of this Peer Review File have been redacted as indicated to remove third-party material where no permission to publish could be obtained.

REVIEWER COMMENTS

Reviewer #1 (Remarks to the Author):

The manuscript analyses the potential for switching fossil fuel workers from fossil fuel jobs to green jobs in the US. The manuscript includes four major aspects of analysis (here I group the authors' second and third points because they address the same question): (1) comparing skill requirements of fossil fuel workers to those required for green jobs; (2) examining if fossil fuel workers are co-located with green job locations now or in the future; (3) analysing historical mobility for fossil fuel workers; and (4) testing the effect of targeted investments on future employment.

The analysis is interesting and policy relevant and the first step in this analysis is particularly novel. At the same time, the manuscript would be improved from additional robustness testing, better situating the work within existing literature, and better explaining the method and reasoning behind the method (and their locations). In general, the manuscript would benefit from explaining all the methods better and significantly expanding the explanation of the methods. In some places I was not able to evaluate the analysis because I could not tell what the authors did from the manuscript. The methods at the moment are about 600 words but I believe the format allows 300 words. Below I structure my review by the major aspects of the analysis and within each I note the issues which arise for each one. Please note that within some of the points, there are sub-items as well.

The most novel aspect of the work is the skill comparison between fossil fuel industries and renewable jobs. At the same time, this aspect of the work needs to be better explained and more work needs to be done to justify that skill similarity is actually larger than distance.

Regarding explanation of skill comparison, it would be useful if the authors can pedagogically explain and justify using the skill similarity data in the way that they do. What types of skills are quantified and what is the robustness of this type of quantification? Here, it would be useful to provide some concrete examples and also refer to literature that shows that using these data is useful for such an analysis.

It would also be good to analyse the skill similarity with other industries and fossil fuel jobs. As the authors rightly note, while there is political rhetoric around fossil fuel to green jobs, there's no clear reason why fossil fuel workers should switch to green jobs and not other sectors. For example, there have been attempts to switch fossil fuel workers to programming. I'm not saying programming should be the object of analysis, but if as the authors argue green jobs are not a plausible target, what is?

Regarding the comparison of the distance effect to the skill similarity effect, it's not convincing that these two can be compared with the analysis presented. The authors normalise all data for the analysis but how do we know that 1 unit of distance is equivalent to 1 unit of skill similarity? It would be good if the authors can provide additional analysis and reasoning around this conclusion since it is a key conclusion in the manuscript.

The second question the authors ask is are green jobs co-located where fossil fuel jobs are. This is a question which has already been explored in the literature with some of the references the authors cite. However, the manuscript does not build on this work nor are the authors explicit about what questions that literature leaves open. It's unclear what gap this part of the analysis fills.

Additionally, the regression analysis depicted in Fig. 2c and 2d is quite unconvincing. Here the authors use power plants as dependent variables but it's unclear why this would depend on or have any relationship to extraction workers, fossil fuel workers or the labor force. In fact, it's likely that wind would be negatively correlated with the labor force since a bigger labor force is likely found in more populous areas where it's more difficult to build wind.

Another issue with this part of the manuscript is the lack of scenario testing. The authors use a projection from BLS for their main analysis and the Reference scenario from Princeton's energy scenario project. But available clean energy jobs has been shown to be closely associated with the growth of renewable energy (more renewables more green jobs). The authors do not test this for different scenario futures and developments of renewables so it's unclear if their results are robust against this future uncertainty.

Additionally, it seems that the authors presume in this step that green jobs are located where good solar and wind resources are. This assumption is also reflected in their statement in the introduction that it is unlikely for solar technology jobs to be deployed in Pittsburgh because of cloud cover. So do I understand the authors correctly that they are considering deployment and O&M jobs but not manufacturing jobs? This would be problematic given that the literature shows that the largest growth is in manufacturing rather than deployment and O&M jobs.

The third piece of their analysis is analysing mobility of fossil fuel workers. I can't evaluate this because it's unclear to me how the authors did the analysis and I could not identify the description of this step in the Methods.

Finally, the authors analyse targeted support for green transitions. Here, again there's the issue of deployment + O&M jobs versus manufacturing jobs. What does the geotargeting entail and how do the authors evaluate this?

Reviewer #2 (Remarks to the Author):

Nature Communications Review
Manuscript#: NCOMMS-23-07759

Comments to author

This paper showcases a sophisticated statistical analysis of fossil fuel workers' likelihood to transition to green jobs using multiple sources of data, including various projections and cross-validation with historical data. The key results show that very few fossil fuel workers will transition to green jobs, and that while both geographic proximity to green job growth areas as well as skill similarity are both barriers, geography plays the largest role in the models. The significance of these findings is that despite the policy mechanisms and narrative supporting a "just transition" for fossil fuel workers, they are not likely to be able to take advantage of programs that aim to supplant fossil fuels jobs with green jobs.

While the statistical methods the authors use is not in my area of expertise, to my mind they seem appropriate and conducted well (as well as thoroughly explained). The sources of data are high quality.

Essentially, I believe the authors' results suggest that green jobs might not be the answer for fossil fuel workers' displacement – instead, new industries should be brought to extractive areas experiencing economic hardship. I do think they could make this policy implication more prominent in the beginning of the paper and in the title. Also, are there analysis or existing literature the authors could reference to further make this point?

The article is well written, though there are some typos and grammatical issues in the abstract.

Overall, I think with some revision this paper could be suitable for publication - see my questions below.

Questions for the authors:

Why only look at solar and wind plants? Green jobs extend beyond this, including factories that make components of green technologies, including solar pv, wind turbines, electric vehicles and batteries, and so on. Also, are the heavy metals used in solar mined in the US, and is mining of those elements considered a "green job?"

Page 4 – What is a "random forest classifier?"

In the discussion, the authors outline their assumptions, which include that there are unaccounted for social barriers to fossil fuel workers taking up green jobs. I suspect the social barriers, including identity and culture, would be hugely consequential reasons for why workers would not transition – I think the authors could tie this idea to the literature more. See the following (Bell and York 2010; Olson-Hazboun 2018; Olson-Hazboun, Howe, and Leiserowitz 2018; York and Bell 2019):

Bell, Shannon Elizabeth, and Richard York. 2010. "Community Economic Identity: The Coal Industry and Ideology Construction in West Virginia: Community Economic Identity." *Rural Sociology* 75(1):111–43. doi: 10.1111/j.1549-0831.2009.00004.x.

Olson-Hazboun, Shawn K. 2018. "'Why Are We Being Punished and They Are Being Rewarded?' Views on Renewable Energy in Fossil Fuels-Based Communities of the U.S. West." *The Extractive Industries and Society* 5(3):366–74. doi: 10.1016/j.exis.2018.05.001.

Olson-Hazboun, Shawn K., Peter D. Howe, and Anthony Leiserowitz. 2018. "The Influence of Extractive Activities on Public Support for Renewable Energy Policy." *Energy Policy* 123:117–26. doi: 10.1016/j.enpol.2018.08.044.

York, Richard, and Shannon Elizabeth Bell. 2019. "Energy Transitions or Additions?: Why a Transition from Fossil Fuels Requires More than the Growth of Renewable Energy." *Energy Research & Social Science* 51:40–43. doi: 10.1016/j.erss.2019.01.008.

Reviewer #3 (Remarks to the Author):

This is a very valuable contribution in terms of exploring the alignment of fossil fuel workers location and renewable energy location. I do have some comments that, I hope, will contribute to improving the paper.

1. It would be useful to disaggregate the number of fossil fuel workers. Does this number include all fossil fuels and does it include extraction, transportation, transformation, transmission, of energy and so on? Does it include 'auxiliary' workers as well as contract workers? This, in my research, is very important. I am not suggesting using industry numbers, however, which are exaggerated. In the case of Colorado, for instance, the JT policy includes extraction, transportation and energy production, at the least.
2. Same with green jobs. What do they include, e.g., EV manufacturing, battery manufacturing....
3. The disaggregation is also important because oil and gas extraction workers tend to be more itinerant than coal workers. And both coal plant and refinery workers are less mobile. If that is so, what are the implications?
4. Your approach to JT is somewhat narrow and akin to economic development. I would recommend that you explore the literature, particularly that which relates to workers and JT more closely. None of the JT references you use deal with workers. Some deal with energy transition and some explore JT with little or partially informed understanding of JT. Only Pai et al., address workers and JT and, even then, in terms of the polyvalent approach to energy justice.

5. JT proposals (from the very beginning) recognized the centrality of relocation, employment/pension security, schooling and retraining, benefits and a green industrial policy. They realized that the places of sunseting and emerging employment did not always align. The alignment of fossil fuel employment with renewables employment is one solution but not the only and probably not even the most likely solution. People may choose other jobs, e.g., manufacturing or working in the service sector doing technical jobs. For that you need some intentionality in your JT policy. Otherwise it is better to use another term, such as labor market dynamics or economic redeployment.

Response to Reviewers:

We thank the reviewers for their extensive and thoughtful assessment of our work “**Quantified Barriers to a Just Transition for US Fossil Fuel Workers**” submitted for consideration for publication with *Nature Communications* (NCOMMS-23-07759). In the following, we address each point raised by the reviewers. Reviewers’ comments are presented in black font while authors’ replies are provided in blue font. We specify each change made to the main text and supplementary materials in the revised submission. We were able to address all comments and we believe our study is now much improved.

Reviewer #1 (Remarks to the Author):

The manuscript analyses the potential for switching fossil fuel workers from fossil fuel jobs to green jobs in the US. The manuscript includes four major aspects of analysis (here I group the authors’ second and third points because they address the same question): (1) comparing skill requirements of fossil fuel workers to those required for green jobs; (2) examining if fossil fuel workers are co-located with green job locations now or in the future; (3) analysing historical mobility for fossil fuel workers; and (4) testing the effect of targeted investments on future employment.

The analysis is interesting and policy relevant and the first step in this analysis is particularly novel. At the same time, the manuscript would be improved from additional robustness testing, better situating the work within existing literature, and better explaining the method and reasoning behind the method (and their locations). In general, the manuscript would benefit from explaining all the methods better and significantly expanding the explanation of the methods. In some places I was not able to evaluate the analysis because I could not tell what the authors did from the manuscript. The methods at the moment are about 600 words but I believe the format allows 300 words. Below I structure my review by the major aspects of the analysis and within each I note the issues which arise for each one. Please note that within some of the points, there are sub-items as well.

Authors: We agree. We have revised the Methods section in the main text to include many more details about both our original analysis and the new analysis added during revision. The revised Methods Section now describes our occupation skill profiles, our method for measuring skill similarity, our job transition and employment data sets, descriptions of our job transition model, and descriptions of our employment projections. We believe this significantly improves the readability of our manuscript.

The most novel aspect of the work is the skill comparison between fossil fuel industries and renewable jobs. At the same time, this aspect of the work needs to be better explained and more work needs to be done to justify that skill similarity is actually larger than distance.

Authors: We thank the reviewer for all of their questions and comments. We were able to clarify all of the points of uncertainty. And we produced several new analyses as a result of their comments. We believe the revised manuscript is much improved as a result.

Regarding explanation of skill comparison, it would be useful if the authors can pedagogically explain and justify using the skill similarity data in the way that they do. What types of skills are quantified and what is the robustness of this type of quantification? Here, it would be useful to provide some concrete examples and also refer to literature that shows that using these data is useful for such an analysis.

Authors: We thank the reviewer raising this clarifying question. We agree that our original submission was not clear on the source or quality of our occupation skill profiles. We are using occupation skill profiles from the O*NET database which is produced by the US Bureau of Labor Statistics (BLS). For over 700 unique occupations, the BLS provides an annual database detailing the real-valued importance of each of 232 O*NET variables to each occupation. Occupation skill profiles are determined according to both surveys of actual workers and expert assessments from BLS analysts. This database is widely used by the US Department of Labor, policy makers, and researchers to study skills in the US workforce. For example, an economics study from the OECD used O*NET data to estimate automation from technology [see Arntz et al, 2016]. As another example, Dierdorf et al (2009) use O*NET data in a US Department of Labor report to identify green occupations. To provide more context for these O*NET skill elements, we have revised the Supplementary Materials with a new section detailing the most important skills for fossil fuel occupations and for green occupations (see Tables S4, S5, S6, and S7).

For convenience, we provide those tables here. First, we report O*NET skills sorted from most to least important to fossil fuel occupations or green occupations, respectively. However, some skills are ubiquitous and required by many different occupations. Therefore, following methods from Alabdulkareem et al (reference [1] in the manuscript), we use location quotient (LQ) to identify the skills that most strongly identify each group of occupations. We also provide tables listing O*NET skills according to their location quotient:

O*NET Skill	Description	Value	LQ
Realistic	Designing, building, or repairing of equipment, engaging in physical activity	1.00	1.57
Handling and Moving Objects	Using hands and arms in handling, installing, positioning, and moving materials	0.83	1.64
Performing General Physical Activities	Performing general physical activities	0.77	1.69
Controlling Machines/Processes	Using control mechanisms or physical activity to operate machines (excluding computers, vehicles)	0.77	1.82
Inspecting Equipment/Structures/Material	Inspecting equipment, structures, or materials to identify the cause of errors	0.76	1.57
Monitor Processes/Materials/Surroundings	Monitoring and reviewing information from materials, events, or the environment	0.69	1.15
Communicating w Supervisors/Peers/Subordinates	Providing information to supervisors, co-workers, and subordinates	0.69	1.03
Identifying Objects/Actions/Events	Identifying information by categorizing, and recognizing differences or similarities	0.67	1.05
Mechanical	Having knowledge of machines and tools, including their designs, uses, repair, maintenance.	0.66	1.91
Repairing/Maintaining Mechanical Equipment	Servicing, repairing, adjusting, and testing machines devices, and moving parts	0.64	2.16

Table 4: Top 10 most important skills for fossil fuel (extraction) workers. Value refers to the O*NET based score that indicates the importance of each skill for extraction workers. This ranges from 0 to 1, where 1 refers to the most important skill. The above 10 skills are skills with the highest O*NET values for fossil fuel (extraction) workers. Location quotient (LQ) measures the relative importance of each skill for extraction workers in comparison to its importance for other occupations. If LQ of a skill is higher than 1, this skill is more important to fossil fuel (extraction) workers than the average occupation.

O*NET Skill	Description	Value	LQ
Sound Localization	to tell the direction from which a sound originated	0.32	3.24
Night Vision	to see under low-light conditions	0.27	3.15
Repairing	to repair machines or systems using the needed tools	0.47	2.88
Glare Sensitivity	to see objects in the presence of a glare or bright lighting.	0.33	2.86
Peripheral Vision	to see objects or movement of objects to one's side	0.27	2.77
Equipment Maintenance	to perform routine maintenance on equipment	0.47	2.76
Speed of Limb Movement	to quickly move the arms and legs	0.36	2.57
Spatial Orientation	to know your location in relation to the environment	0.34	2.52
Installation	to install equipment, machines, wiring, or programs to meet specifications	0.14	2.50
Gross Body Equilibrium	to keep/regain your body balance or stay upright in an unstable position.	0.42	2.44

Table 5: Top 10 skills with relative importance for fossil fuel (extraction) workers. The above 10 skills are skills with the highest location quotient (LQ) for fossil-fuel (extraction) workers. LQ measures the relative importance of each skill for extraction workers in comparison to its importance for other occupations. If LQ of a skill is higher than 1, this skill is more important to fossil fuel (extraction) workers than the average occupation.

O*NET Skill	Description	Value	LQ
Realistic	Designing, building, or repairing of equipment, engaging in physical activity	0.83	1.31
Making Decisions and Solving Problems	Analyzing information and evaluating results	0.74	1.11
Getting Information	Observing, receiving, and otherwise obtaining information from all relevant sources	0.73	1.06
Communicating with Supervisors, Peers, or Subordinates	Providing information to co-workers	0.72	1.08
Updating/Using Relevant Knowledge	Keeping up-to-date technically and applying new knowledge to your job	0.71	1.09
Identifying Objects, Actions, Events	Identifying information by categorizing, estimating, recognizing differences or similarities	0.71	1.11
Monitor Processes, Materials, or Surroundings	Monitoring and reviewing information from materials, events, or the environment	0.71	1.18
Evaluating Information to Determine Compliance with Standards	Using information and judgment to determine whether events or processes comply with laws or standards	0.68	1.22
Processing Information	Compiling, coding, categorizing, calculating, tabulating, auditing, or verifying information or data	0.68	1.14
Inspecting Equipment, Structures, Material	Inspecting equipment, structures, or materials to identify the cause of errors	0.67	1.37

Table 6: Top 10 most important skills for green jobs in the renewable energy sector. Value refers to the O*NET based score that indicates the importance of each skill for green occupations in renewable energy generation sector. This ranges from 0 to 1, where 1 refers to the most important skill. The above 10 skills are skills with the highest O*NET values for green occupations in renewable energy generation sector. Location quotient (LQ) measures the relative importance of each skill for green occupations in comparison to its importance for other occupations. If LQ of a skill is higher than 1, this skill is more important to fossil fuel (extraction) workers than the average occupation.

O*NET Skill	Description	Value	LQ
Physics	Having knowledge of physical principles	0.51	2.18
Repairing	Repairing machines using the needed tools	0.32	1.95
Engineering and Technology	Having knowledge of the design, development, and application of technology for specific purposes	0.60	1.92
Drafting, Laying Out, Specifying Technical Devices	Providing documentation, detailed instructions, to tell others about how devices, parts, equipment are to be fabricated	0.46	1.86
Equipment Maintenance	Performing routine maintenance on equipment and determining when and what kind of maintenance is needed.	0.30	1.78
Troubleshooting	Determining causes of operating errors and solutions	0.43	1.78
Science	Using scientific rules and methods to solve problems	0.36	1.72
Repairing and Maintaining Electronic Equipment	Servicing, repairing, calibrating, regulating, fine-tuning, or testing machines, devices, and equipment	0.45	1.71
Design	Having knowledge of design techniques, tools, and principles involved in production of precision technical plans.	0.48	1.69
Mechanical	Having knowledge of machines and tools, including their designs, uses, repair, and maintenance.	0.56	1.61

Table 7: Top 10 skills with relative importance for green occupations in the renewable energy sector. The above 10 skills are skills with the highest location quotient (LQ) for green occupations.

We have updated our description of the skills data in the Methods section to the content below and referenced this material in our Results section:

*We use occupation skill profiles from the O*NET database. O*NET is an annually updated database produced and maintained by the US Bureau of Labor Statistics (BLS). The BLS produces these skill profiles using surveys of actual workers and assessments from BLS analysts. O*NET provides information on required workplace*

*skills for over 750 occupations defined by Standard Occupation Classification (SOC) taxonomy. The BLS uses SOC codes to also report employment distributions for regions (e.g., MSAs, NMSAs, states, and nationally). O*NET identifies the 232 different skills and rates their importance in each occupation in a continuous range of 0 to 1, where 1 is the most important skill for a given occupation. O*NET data is widely used by the US Department of Labor, policy makers, and researchers to study skills in the US workforce. For example, an economics study from the OECD used O*NET data to estimate automation from technology [4]. As another example, O*NET data is used in a US Department of Labor report to identify green occupations [11]. Tables S3, S4, S5, and S6 detail the most important skills for fossil fuel occupations and green industry occupations.*

It would also be good to analyse the skill similarity with other industries and fossil fuel jobs. As the authors rightly note, while there is political rhetoric around fossil fuel to green jobs, there's no clear reason why fossil fuel workers should switch to green jobs and not other sectors. For example, there have been attempts to switch fossil fuel workers to programming. I'm not saying programming should be the object of analysis, but if as the authors argue green jobs are not a plausible target, what is?

Authors: We thank the reviewer for this suggestion. Our original analysis did compare the similarity of fossil fuel workers' skills to the skills required by green occupations and other occupations. Figure 1A shows that, on average, fossil fuel workers' skills are more similar to the skills required by green occupations than to the skills required by other occupations. From a purely skills-based perspective, this suggests that fossil fuel workers may more easily transition to green occupations than to other occupations because relatively less re-skilling would be required.

We have created a new Figure and Table in the SI detailing the skill similarity between fossil fuel workers and each industry in the US economy. SI Section 3.2 and SI Section 3.3. Figure 9 from SI section 3.2 shows pairwise skill similarities across industries (defined by NAICS 2 digit) including skill similarity score between fossil fuel jobs and green jobs. Table 8 from section 3.3 reports the skill similarities between fossil fuel industry and other industries (defined by NAICS 2 digit). For convenience, we provide screenshots of these results here:

Figure 9: Distribution of pair-wise skill similarities across industries (NAICS). The mean of pair-wise skill similarities = 0.82 (red line) and the skill similarity score between fossil-fuel industry and green jobs = 0.91 (green line).

NAICS	Description	Similarity
23	Construction	0.94
22	Utilities	0.89
48-49	Transportation and Warehousing	0.88
81	Other Services (except Public Administration)	0.85
31-33	Manufacturing	0.85
53	Real Estate and Rental and Leasing	0.84
42	Wholesale Trade	0.84
56	Administrative and Support and Waste Management and Remediation Services	0.83
71	Arts, Entertainment, and Recreation	0.81
62	Health Care and Social Assistance	0.80
44-45	Retail Trade	0.80
11	Agriculture, Forestry, Fishing and Hunting	0.79
51	Information	0.79
72	Accommodation and Food Services	0.79
54	Professional, Scientific, and Technical Services	0.76
55	Management of Companies and Enterprises	0.76
61	Educational Services	0.74
52	Finance and Insurance	0.72

Table 8: Skill similarities between Fossil-fuel workers and other industries (NAICS)

Regarding the comparison of the distance effect to the skill similarity effect, it's not convincing that these two can be compared with the analysis presented. The authors normalise all data for the analysis but how do we know that 1 unit of distance is equivalent to 1 unit of skill similarity? It would be good if the authors can provide

additional analysis and reasoning around this conclusion since it is a key conclusion in the manuscript.

Authors: We agree normalizing skill similarity and distance to be between 0 and 1 would indeed make the regression coefficients difficult to compare. However, we actually normalized variables to be z-scores; that is, for each variable, we subtract the mean and then divide by the standard deviation. This normalization is performed for the dependent variable as well. Then, the regression coefficients describe how a 1 standard deviation change in an independent variable corresponds to standard deviation changes in the dependent variable. This approach enables us to compare the relationships between variables and mobility directly because z-scores are unitless and respect the naturally occurring variation in each variable.

Our main goal is to compare the importance of skill similarity to the importance of distance. Our skill similarity metric is unitless by construction and ranges from 0 to 1, but it is not clear beforehand what ranges of skill similarity scores we might actually observe in the data (e.g., it's not clear if a skill similarity score of 0.50 is relatively large or small). At the same time, distance has units of measure (i.e., miles) and offers the same challenge of identifying large or small distances. We directly convert the skill similarity variables to z-scores, and we also convert the distance variables to a z-score after a logarithmic transformation. Converting both variables to z-scores makes them unitless and removes the need to subjectively interpret what a large or small change in that variable might be; instead, we compare changes in standard deviation according to the regression coefficients.

For example, using Model 5 in Figure 1B, we find that a standard deviation increase in skill similarity corresponds to the 0.41 standard deviation increase in worker transitions from fossil fuel occupations to occupations in other industries. However, a standard deviation increase in distance corresponds to a 2.07 standard deviation decrease in transitions. In absolute terms, the standard deviation change in distance is associated with a larger change in transitions than a standard deviation increase in skill similarity. And so, we conclude that distance has a larger impact on transitions than skill similarity.

Our original submission was not clear about our normalization process. We have updated the caption of Figure 1 to the following:

Although both are significant, geospatial distance is a bigger factor than skill similarity in fossil fuel worker mobility. (A) Fossil fuel workers' skills are more similar to the skill requirements of green jobs than to those of other industries according to a two-sample t-test. (B) We use a Poisson model to predict the flows of workers who transition from

the fossil fuel industry (f) in metropolitan area m to industry i in metropolitan area m' according to industry-region migration data from the US Census Bureau from 2005 to 2019. Distance and employment are log-transformed. The Stay (Industry) and Stay (Location) indicator variables control for workers who remain in the same industry or MSA/NMSA, respectively. All variables are centered and standardized (i.e., transformed into z-scores) so regression coefficients are directly comparable.

We have also revised the paragraph describing Figure 1 in the Results section to include the following:

All variables are centered and standardized (i.e., transformed to z-scores) to remove units so that coefficient estimates are comparable across variables (i.e., coefficients represent changes in standard deviations). This normalization enables us to identify how a standard deviation change in a variable (e.g., skill similarity) corresponds to standard deviation changes in the worker flows from fossil fuel occupations to other industries, and, in particular, it allows us to compare which variables are most strongly associated with fossil fuel worker mobility relative to each variable's natural variability.

The second question the authors ask is are green jobs co-located where fossil fuel jobs are. This is a question which has already been explored in the literature with some of the references the authors cite. However, the manuscript does not build on this work nor are the authors explicit about what questions that literature leaves open. It's unclear what gap this part of the analysis fills.

Authors: Indeed, we should have clarified our contributions and connections to the extant literature. We try to address this concern in two ways. First, we completed another round of literature review allowing us to include relevant studies that we might have missed. Second, we edited the *Introduction* and the *Discussion* sections to clarify our contribution with regards to the co-location problem.

To the best of our knowledge, one of the most systematic and rigorous studies on geographical co-location remains Pai et al. (2020), who themselves build on earlier studies (mostly about the UK) that show low geographical mobility of coal workers (e.g., Hollywood 2002, Beatty et al. 2007). Our paper departs from theirs in several significant ways. First, they focus on coal miners (on the fossil fuel side) and on solar and wind jobs (on the green side). We take a more disaggregated view of both sides, looking at a wide portfolio of existing and potential occupations. Second, Pai et al. (2020) focus on workers located near coal mines. This makes sense given their focus on coal mining jobs. Our approach, however, is more flexible and acknowledges that jobs in the fossil

fuel sector can be located outside of mining and digging regions. Second, their paper asks what would happen if (most if not all) coal miners were immobile. We use US Bureau of Labor Statistics data to relax this assumption and model what could happen given historical mobility patterns of fossil fuel workers.

Another related study is Vanatta et al. (2022). This study models the deployment of renewable energy to replace coal jobs in the United States. Their research question resembles Pai et al.'s: can wind and solar jobs be created near coal mines to offer alternative livelihoods to coal workers? As before, our analysis departs from theirs by relaxing the assumption of no/little geographical movement. Instead, we let the data speak on this particular dimension.

Concretely, we added the following mention:

The problem of geographical co-location between fossil and green jobs has started to receive more attention, but existing studies typically do not examine the broader portfolio of occupations, nor do they estimate the mobility of fossil fuel workers (Pai et al. 2020, Vanatta et al. 2022).

Additionally, the regression analysis depicted in Fig. 2c and 2d is quite unconvincing. Here the authors use power plants as dependent variables but it's unclear why this would depend on or have any relationship to extraction workers, fossil fuel workers or the labor force. In fact, it's likely that wind would be negatively correlated with the labor force since a bigger labor force is likely found in more populous areas where it's more difficult to build wind.

Authors: We agree completely. This regression is meant to be purely correlational and we in no way want to suggest that fossil fuel worker employment causes green energy power plants. Our goal is to demonstrate the correlation between the distribution of fossil fuel workers and green power plants while controlling for the size of the local labor market. Our analysis reveals a negative or weakly positive correlation between fossil fuel workers and power plants. This is another piece of evidence supporting our claim that distance is a barrier to a transition from fossil fuel jobs to green industry jobs, at least to the extent that the locations of current green power plants indicate green employment.

Our explanation of this analysis was not clear in our original submission. We have revised the caption to Figure 2D to read:

(D) Poisson regressions investigating the 2019 correlation between fossil fuel worker employment and the number of green energy plants while controlling for the size of the local labor market.

We have also revised our description of Figure 2 in the Results section to avoid future confusion:

Since distance has been a barrier to fossil fuel workers' mobility, are today's fossil fuels workers co-located with green jobs? Investigating this question is challenging. While green employment is expected to grow over the next decade, it is hard to forecast where new jobs will appear. Therefore, we first consider the locations of today's green energy producing power plants using data from the US Energy Information Administration to approximate the labor markets currently supporting green jobs. In Figure 2 A&B, we map the locations of today's solar and wind power plants and compare them to the 2019 employment distribution (i.e., before the COVID pandemic) of fossil fuel workers in MSAs and NMSAs (see SI Section 5 for a similar analysis of other energy sources). In both cases, this correlational analysis reveals that the number of plants does not strongly correlate with fossil fuel worker employment according to a cross-sectional comparison. Most regions are dominated by only power plants or fossil fuel worker employment, but not both (see Fig. 2C). A Poisson regression that accounts for labor market size reveals only weak associations between the spatial distribution of green energy power plants and fossil fuel workers (see Fig. 2D).

Another issue with this part of the manuscript is the lack of scenario testing. The authors use a projection from BLS for their main analysis and the Reference scenario from Princeton's energy scenario project. But available clean energy jobs has been shown to be closely associated with the growth of renewable energy (more renewables more green jobs). The authors do not test this for different scenario futures and developments of renewables so it's unclear if their results are robust against this future uncertainty.

Authors: We agree with the reviewer that the future distribution of green employment is highly uncertain. To address this, our analysis does consider several different possible scenarios.

First, our analysis considers possible distribution of future green employment according to (1) a model informed by BLS employment projections, (2) state-level estimates from the Princeton energy scenario project, and (3) a conservative estimate of future employment based on the current locations of green energy power plants. Second, our analysis also considers several different definition for "green occupation" including (1) merging occupations in the "renewable energy sector" identified by Dierdorff et al in

their US Department of Labor report (see reference [13] in the revised manuscript), (2) list of all occupations in the renewable energy sector in BLS projections for 2029, (3) occupations in the renewable energy sector (solar, wind, biomass, hydro) that are projected to grow either in the number or share in employment, and (4) all the “green increased demand” jobs defined by [13] beyond occupations in the renewable energy sector. We compare these alternative definitions in the Supplementary Materials Section 2.

Additionally, motivated by the Biden Administration’s Inflation Reduction Act (IRA), we consider potential future scenarios where new employment is incentivized (1) proportional to existing labor market size (i.e., untargeted) or (2) proportional to current fossil fuel worker employment (i.e., targeted). We estimate fossil fuel worker transitions to green industry jobs under these various scenarios in Figure 4C. Our results reveal significantly increased transitions from fossil fuel employment to green employment for far less investment (i.e., less money) in the targeted scenario. For example, we estimate that a targeted investment of \$1 million will produce more transitions from fossil fuel to green jobs than an untargeted investment of \$5 million.

To make these robustness checks clear in the future, we have revised our Discussion section to include the following:

Our modeling approach makes several assumptions that limit our analysis. First, we have assumed that all of today's fossil fuel workers will want to transition to green jobs. However, there may be additional social barriers including workers' preferences, identity, culture, and economic outlook. Second, workers from other industries may compete with fossil fuel workers to fill green industry jobs. Third, we have looked at a narrow subset of extraction workers but other workers in the fossil fuel sector, such as engineers and lawyers, would also need to find new jobs if the sector phased out entirely. Fourth, currently unknown occupations may emerge as the green industry continues to evolve. With these limitations in mind, our results represent a best-case scenario for fossil fuel workers even though there exist additional barriers to a Just Transition. Because of these necessary assumptions, we include several alternative approaches to our analysis as robustness checks; for example, we vary our definition of “green occupation” (see SI Section 2), our estimates of future green employment growth vary from a conservative approach based on the location of current green energy power plants to a more complex approach based on employment predictions from the US Bureau of Labor Statistics, and consider the potential impact of federal stimulus in fossil fuel communities (see Fig. 4C).

Additionally, it seems that the authors presume in this step that green jobs are located where good solar and wind resources are. This assumption is also reflected in their statement in the introduction that it is unlikely for solar technology jobs to be deployed in Pittsburgh because of cloud cover. So do I understand the authors correctly that they are considering deployment and O&M jobs but not manufacturing jobs? This would be problematic given that the literature shows that the largest growth is in manufacturing rather than deployment and O&M jobs.

Authors: We fully agree with the reviewer that fossil fuel workers can transition to other sectors as well. To address this, we perform a new analysis examining the possibility of fossil fuel workers transitioning to different industries, including the manufacturing industry, in three different ways in the revised manuscript and supplementary material.

First, in Figure 4C, we show that fossil-fuel workers' predicted transition to six other sectors: construction, manufacturing, transportation and warehousing, which have the highest skill similarities, and three other industries with the lowest skill similarities with fossil fuel industry. To estimate the fossil fuel workers' transition rates to these industries, we identify the core occupations of each industry using national employment projection data by BLS. Then, using these occupations and the predicted value of their employment in 2029, along with the regression model from Figure 1B, model (5), we project the potential transition from fossil fuel sector to these industries.

We find that the fossil fuel workers' predicted transition rates to the manufacturing and construction sector is around 3%, which is higher than their transition rate to the renewable energy sector, assuming no major policy interventions (baseline scenario in figure 4C). For convenience, we provide screenshots of the results here:

Second, in Supplemental Material, we show the group of occupations that are most likely for fossil fuel workers to transition to, using the major and minor occupation group

defined by Standard Occupation Classification. Consistent to the industry level transition analysis (Figure 4C), Figure S11 shows that manufacturing and construction related occupations are the occupation groups with the highest predicted transition rates for fossil fuel workers in 2029.

Figure 11: 5 major and minor occupation groups with the highest transition rates of fossil-fuel extraction workers

Lastly, we consider fossil-fuel workers' transition to different types of green occupations, including green industry manufacturing opportunities. While we primarily focus on the green occupations associated with renewable energy generation in the main analysis, there are different types of green occupations that are associated with other sectors defined by Dierdorff et al (Detailed discussion on different sectors in the green economy related sectors are provided in the revised Supplementary Material section 2). Dierdorff et al. provides a list of green occupations in 12 different sectors including green transportation, energy trading, and manufacturing. Manufacturing in this context is defined as a sector that “covers activities related to industrial manufacturing of green technology as well as energy efficient manufacturing processes.” While some occupations in this category are not exclusive to the green economy (i.e. Industrial Machinery Mechanics, Iron and Steel Workers, etc), in this analysis we assume all the occupations in each category as green occupations. In Figure S13 in SI, we show the predicted transition rates of fossil fuel workers to 12 different sectors of green occupations.

Figure 13: Transition to different types of green jobs

The third piece of their analysis is analyzing mobility of fossil fuel workers. I can't evaluate this because it's unclear to me how the authors did the analysis and I could not identify the description of this step in the Methods.

Authors: We agree. We have revised two subsections in the Methods with more details about the data sources and calculations required for the mobility analysis. The revised materials includes:

Comparing skill requirements across industries

To estimate skill similarity at the sector level, we use the weighted average of the importance of each skill ($onet(j, s)$) with the share of employment in occupation j in industry i . In other words, we measured the skill similarity between average workers in the fossil-fuel sector and other industry i' . To accomplish this, we represent the importance of skill s to industry i according to

$$onet(i, s) = \frac{\sum_{j \in Jobs} onet_{j,s} \cdot N_{j,i}}{\sum_{j' \in Jobs} N_{j',i}}$$

where $N_{j,i}$ is the employment count for occupation j within industry i .

Then, we calculate the skill similarity between industries i and i' according to

$$\text{Skill similarity}_{i,i'} = \frac{\sum_{s \in S} \min(onet(i, s), onet(i', s))}{\sum_{s \in S} \max(onet(i, s), onet(i', s))}$$

Section 3.3 in the Supplementary Materials provides the distribution of industry-pair skill similarity scores and a list of the US sectors with the greatest skill similarity to fossil fuel occupations. Construction (NAICS 23) and Utilities (NAICS 22) have the greatest skill similarity to fossil fuel occupations.

Modeling job transitions between industries

We estimate variations of the following model with 10-fold cross-validation:

$$\begin{aligned} \text{Transition}_{f,m,i',m'} &\sim \lambda_{f,m,i',m'} \\ \log(\lambda_{f,m,i',m'}) &= \beta_0 + \beta_1 \log(\text{Distance})_{m,m'} + \beta_2 \text{Skill similarity}_{f,i'} \\ &\quad + \beta_3 \text{Employment}_{f,m} + \beta_4 \text{Employment}_{i',m'} \end{aligned} \quad (4)$$

The distance between regions m and m' (i.e., MSAs or NMSAs) is calculated in miles from the centroid of the regions' Tiger Shape File provided by the US Census Bureau. Distances are calculated using the Haversine formula. The dependent variable in this analysis comes from the J2JOD data provided by the US Census Bureau and the employment statistics come from the US Bureau of Labor Statistics.

Finally, the authors analyse targeted support for green transitions. Here, again there's the issue of deployment + O&M jobs versus manufacturing jobs. What does the geotargeting entail and how do the authors evaluate this?

Authors: We thank the reviewer for the question. Please see our reply above for a discussion of green industry manufacturing jobs. In geotargeting scenarios, we consider policy interventions that create green jobs specifically targeted to regions with a heavy reliance on fossil fuels in their labor market. In comparison, in nontargeting scenarios, we assume that the growth rates of green jobs will be proportional to the current employment share in the region. In the revised manuscript, we provide a more detailed explanation in Method section as follow:

Transition to green jobs under different scenarios

For non-targeted scenarios, we increase the number of fossil fuel jobs with the equal rate for every metro and non metropolitan area from our original predicted number of green jobs (baseline). For geo-targeted scenarios, we distribute the green jobs proportional to the share of fossil fuel workers across regions. The number of each green job (i) in each region (m) that is increased by different scenarios can be represented as follows:

$$GOCC_{i,m} + (\# \text{ of Jobs created}) \times \frac{GOCC_{i,m}}{\sum_{i,m} GOCC_{i,m}} \quad (\text{Non-Targeted})$$

$$GOCC_{i,m} + (\# \text{ of Jobs created}) \times \frac{GOCC_{i,m}}{\sum_i GOCC_{i,m}} \times \frac{\sum_i FOCC_{i,m}}{\sum_{i,m} FOCC_{i,m}} \quad (\text{Geo-Targeted})$$

where $GOCC_{i,m}$ as the predicted number of green job i in area m in baseline scenario (without considering any policy interventions), $FOCC_{i,m}$ as the number of fossil-fuel extraction job i in area m in 2019. In our analyses, we vary “# of Jobs created” by 1M, 5M, and 10M. Across these six different scenarios, we assume that the the proportion of different green occupations within the region are the same with our baseline scenario prediction for 2029. In other words, we do not take into account specific occupation based targeting. Using these different scenarios as future green job supplies, we use our transition model to predict the proportion of fossil fuel workers in 2019 that will be able to transition.

Figure S12 shows the different spatial distribution of green jobs by whether policy interventions to create green jobs are geo-targeted.

Figure 12: Distribution of the logged number of green jobs under **A** scenario where 5 million additional green jobs distributed proportional to the current distribution of green jobs (Non-Targeted (5M)), **B** scenario where 5 million additional green jobs distributed proportional to the current distribution of fossil fuel extraction workers. (Geo-Targeted (5M))

Reviewer #2 (Remarks to the Author):

Nature Communications Review
Manuscript#: NCOMMS-23-07759

Comments to author

This paper showcases a sophisticated statistical analysis of fossil fuel workers' likelihood to transition to green jobs using multiple sources of data, including various projections and cross-validation with historical data. The key results show that very few fossil fuel workers will transition to green jobs, and that while both geographic proximity to green job growth areas as well as skill similarity are both barriers, geography plays the largest role in the models. The significance of these findings is that despite the policy mechanisms and narrative supporting a "just transition" for fossil fuel workers, they are not likely to be able to take advantage of programs that aim to supplant fossil fuels jobs with green jobs.

While the statistical methods the authors use is not in my area of expertise, to my mind they seem appropriate and conducted well (as well as thoroughly explained). The sources of data are high quality.

Essentially, I believe the authors' results suggest that green jobs might not be the answer for fossil fuel workers' displacement – instead, new industries should be brought to extractive areas experiencing economic hardship. I do think they could make this policy implication more prominent in the beginning of the paper and in the title. Also, are there analysis or existing literature the authors could reference to further make this point?

Authors: We agree with the reviewer. We revised the last paragraph of the Introduction to include the following:

Our findings demonstrate that today's fossil fuel workers are, mostly, appropriately skilled for green occupations but are not located in the regions where green jobs are likely to emerge. Thus, prudent policy in support of a Just Transition must address this co-location issue either by creating incentives for today's fossil fuel workers to relocate or by stimulating new employment opportunities in the regions where fossil fuel workers currently reside.

The article is well written, though there are some typos and grammatical issues in the abstract.

Overall, I think with some revision this paper could be suitable for publication - see my questions below.

Questions for the authors:

Why only look at solar and wind plants? Green jobs extend beyond this, including factories that make components of green technologies, including solar pv, wind turbines, electric vehicles and batteries, and so on. Also, are the heavy metals used in solar mined in the US, and is mining of those elements considered a “green job?”

Authors: We thank the reviewer for this question. Our original analysis focused on only solar and wind plants in the main text, but also included additional analyses of hydro and biomass power plants in Section 5 of the Supplementary Materials. We have revised the caption of Figure 2 with a pointer to this SI section to avoid this confusion in the future.

Additionally, we have revised our manuscript with new analysis estimating fossil fuel worker transitions to other industries (i.e., NAICS codes) besides green industry occupations. For example, in revised Figure 4C, we find that fossil fuel worker transitions to Manufacturing jobs will be comparable to transitions to green occupations in the absence of policy interventions that create new green jobs beyond employment projections from the US Bureau of Labor Statistics. We provide Figure 4C here for convenience:

Additionally, we consider fossil-fuel workers’ transition to different types of green occupations, including green industry manufacturing opportunities. While we primarily focus on the green occupations associated with renewable energy generation in the main analysis, there are different types of green occupations that are associated with other sectors defined by Dierdorff et al [13]. Dierdorff et al. provides a list of green occupations in 12 different sectors including green transportation, energy trading, and manufacturing. Manufacturing in this context is defined as a sector that “covers activities related to industrial manufacturing of green technology as well as energy efficient manufacturing processes.” While some occupations in this category are not exclusive to green sectors (i.e.Industrial Machinery Mechanics, Iron and Steel Workers, etc), in this analysis we assume all the occupations in each category as green occupations. In

Figure S13 in SI, we show the predicted transition rates of fossil fuel workers to 12 different sectors of green occupations.

Figure 13: Transition to different types of green jobs

Page 4 – What is a “random forest classifier?”

Authors: This is actually a typo in our original manuscript; in fact, we use “random forest regression” in our analysis rather than a classifier. We have revised the manuscript accordingly.

A random forest is a meta estimator that fits a number of classifying decision trees on various sub-samples of the dataset and uses averaging to improve the predictive accuracy and control over-fitting. The origins of this method date back nearly 30 years to Ho, Tin Kam. "Random decision forests." Proceedings of 3rd international conference on document analysis and recognition. Vol. 1. IEEE, 1995. However, our implementation uses more modern versions of this method supported through the Scikit-Learn Python package (link). References for this implementation are provided below and we have added them to the revised manuscript. Random forest regressors are extremely predictive models because they allow nonlinear interactions between independent variables and they implicitly combat overfitting to the training data. In our study, we demonstrate the predictive performance of the random forest regression out-of-sample prediction performance according to 10-fold cross-validation.

- Breiman, “Random Forests”, Machine Learning, 45(1), 5-32, 2001.
- P. Geurts, D. Ernst., and L. Wehenkel, “Extremely randomized trees”, Machine Learning, 63(1), 3-42, 2006.

- Ho, Tin Kam. "Random decision forests." Proceedings of 3rd international conference on document analysis and recognition. Vol. 1. IEEE, 1995.

In the discussion, the authors outline their assumptions, which include that there are unaccounted for social barriers to fossil fuel workers taking up green jobs. I suspect the social barriers, including identity and culture, would be hugely consequential reasons for why workers would not transition – I think the authors could tie this idea to the literature more. See the following (Bell and York 2010; Olson-Hazboun 2018; Olson-Hazboun, Howe, and Leiserowitz 2018; York and Bell 2019):

Bell, Shannon Elizabeth, and Richard York. 2010. "Community Economic Identity: The Coal Industry and Ideology Construction in West Virginia: Community Economic Identity." *Rural Sociology* 75(1):111–43. doi: 10.1111/j.1549-0831.2009.00004.x.

Olson-Hazboun, Shawn K. 2018. "Why Are We Being Punished and They Are Being Rewarded?' Views on Renewable Energy in Fossil Fuels-Based Communities of the U.S. West." *The Extractive Industries and Society* 5(3):366–74. doi: 10.1016/j.exis.2018.05.001.

Olson-Hazboun, Shawn K., Peter D. Howe, and Anthony Leiserowitz. 2018. "The Influence of Extractive Activities on Public Support for Renewable Energy Policy." *Energy Policy* 123:117–26. doi: 10.1016/j.enpol.2018.08.044.

York, Richard, and Shannon Elizabeth Bell. 2019. "Energy Transitions or Additions?: Why a Transition from Fossil Fuels Requires More than the Growth of Renewable Energy." *Energy Research & Social Science* 51:40–43. doi: 10.1016/j.erss.2019.01.008.

Authors: We thank the reviewer for suggesting these references. We have revised our Discussion section to include citations to these references when discussing potential social barriers.

Reviewer #3 (Remarks to the Author):

This is a very valuable contribution in terms of exploring the alignment of fossil fuel workers location and renewable energy location. I do have some comments that, I hope, will contribute to improving the paper.

1. It would be useful to disaggregate the number of fossil fuel workers. Does this number include all fossil fuels and does it include extraction, transportation, transformation, transmission, of energy and so on? Does it include 'auxiliary' workers as well as contract workers? This, in my research, is very important. I am not suggesting using industry numbers, however, which are exaggerated. In the case of Colorado, for

instance, the JT policy includes extraction, transportation and energy production, at the least.

Authors: We thank the reviewer for this clarifying question. We consider a few different definitions for fossil fuel occupations in our study. Our results are robust across several definitions.

In the Supplementary Material, we include more detailed explanation on our approaches to identify the fossil fuel workers as follows:

1.1. Identifying Fossil Fuel Workers

First, for the main analysis, we focus on 11 occupations that fall into the “extraction workers” category by Standard Occupation Classification (SOC) code by BLS. We focus on these occupations for two reasons. First, they are well represented in the fossil fuel sector: about 70% of extraction workers are active in fossil fuel firms (as of 2019). Second, they represent a core of the fossil fuel business (representing about 27% of the workers in the industry). The next largest groups within fossil fuel firms – engineers and material movers – represent much smaller shares (about 13% and 10% respectively) of fossil fuel workers and have clear outside options. As such, we use a narrow definition of fossil fuel workers in order to focus on the segment that is the most at risk in the clean energy transition. From BLS employment data, we use the number of workers with extraction occupations in metro and non-metropolitan areas where fossil fuel is extracted and use them as a proxy for the number of fossil fuel workers. In other words, we exclude workers with extraction occupations but work in areas without any fossil fuel extraction. To identify areas with active fossil fuel extraction activities, we use county-level oil, gas, and coal production data sourced from the U.S. Department of Agriculture.

[redacted]

1.2. Alternative Measure of Fossil Fuel Workers

Second, while we focus on extraction workers as the core workforce in the fossil fuel industry, there are broader sets of occupations present in the fossil fuel industry. As an alternative, we focus on 40 occupations present within the fossil fuel industry that require manual labor as their main tasks. This includes all the extraction occupations and other occupations in “Installation, Maintenance, and Repair Occupations (SOC code 49-0000)”, “Production Occupations (SOC code 51-000)”, and some of the occupations in “Transportation and Material Moving Occupations (53-0000).” We use the number of workers with manual occupations within the fossil fuel industry in metro and nonmetropolitan areas where fossil fuel is extracted.

Using the fossil fuel manual workforce as an alternative measure, we predict the number of manual workers who will be able to transition to green jobs. Using this alternative measure for fossil fuel workers, we get similar results as the analysis in the main text: Proportion of fossil fuel workers who transition to green jobs in 15 most fossil fuel labor-intensive regions remains lower than 1.1 % (Figure S3) and the proportion of entire fossil fuel workers who transition to green occupation is about 0.78% which is lower than baseline predicted transition rates (1.03 %)

Figure 3: (A) Proportion of fossil fuel who transition to green jobs in 15 most fossil fuel labor-intensive regions remains lower than 1.15 % when we focus on all the manual labor workers in the fossil fuel industry instead of focusing on extraction workers. (B) Within the workers who transition to green jobs, the share of workers who relocate for new green jobs remains low as well.

2. Same with green jobs. What do they include, e.g., EV manufacturing, battery manufacturing....

Authors: We thank the reviewer for this suggestion. In the revised supplementary material section 2.1, we provide the full list of green occupations used in our primary analysis as well as more detailed explanations on how we identify green occupations and alternative ways.

First, in the main analysis, we focus on 24 occupations that are associated with the renewable energy generation sector, defined by Dierdorff et al. Green occupations in the renewable energy generation sector are occupations that are related to developing and using renewable energy sources such as wind, solar, and hydropower. This includes occupations such as 'solar photovoltaic installers', 'wind turbine service technicians', 'power plant operators', among others. In the SI 2.1, we list all the green occupations in the renewable energy generation sector. For convenience, we provide a screenshot of the table.

SOC	Type	Occupation Title
47-2231	GNE	Solar photovoltaic installers
49-9081	GNE	Wind turbine service technicians
49-9099	GNE	Installation, maintenance, and repair workers, all other
47-4099	GNE	Miscellaneous construction and related workers*
47-1011	GNE	First-line supervisors of construction trades and extraction workers
41-4011	GNE	Sales representatives, technical and scientific products*
47-2211	GES	Sheet metal workers*
49-9042	GES	Maintenance and Repair Workers, General
51-9012	GES	Separating, filtering.. and still machine setters, operators*
51-8099	GNE	Plant and system operators, all other
51-8013	GES	Power plant operators*
51-8012	GID	Power distributors and dispatchers
51-8011	GES	Nuclear power reactor operators
51-4041	GES	Machinists
19-4041	GES	Geological and hydrologic technicians*
19-4051	GES	Nuclear technicians
17-2051	GES	Civil engineers*
17-2071	GES	Electrical engineers*
17-2141	GES	Mechanical engineers*
17-2199	GNE	Engineers, all other*
11-3051	GNE	Industrial production managers*
11-3071	GES	Transportation, storage, and distribution managers*
11-9041	GNE	Architectural and engineering managers*
11-9199	GNE	Managers, all other*

Table 2: List of green occupations in Renewable Energy Sector. SOC reports the Standard Occupation Classification 6 digit code for each occupation. Occupations with * are associated different green sectors. For instance, Sheet metal workers (47-2211) are associated with green construction and manufacturing as well by Dierdorff et al [2].

Also, we added the detailed explanation on the definitions of different green jobs as well as justification of using renewable energy generation occupations as core green jobs in our primary analysis in the Supplementary Material section 2.1. For convenience, we provide part of section 2.1. here:

Dierdorff and colleagues \cite{Dierdorffetal2009} categorize the green occupations in the following 12 different green economy related sub-sectors... In the main analyses, we use the 24 occupations that are identified as occupations associated with the Renewable Energy Generation sector. The Renewable Energy Generation sector primarily includes occupations that are related to developing and using renewable energy sources (wind, solar, hydropower, hydrogen, biomass and geothermal).

We focus on Renewable Energy Generation sector occupations as core green occupations because occupations in Renewable Energy Generation are considered to be more closely associated with the green economy compared to other sectors. One challenge of identifying green jobs is that many green occupations defined by Dierdorff are not exclusively confined to the green economy. For example, bus drivers are classified as GID occupations within the green transportation sector, which

encompasses `activities related to increasing efficiency and/or reducing environmental impact of various modes of transportation." While the demand for bus drivers is expected to increase as the green transportation sector expands, the proportion of green transportation employment within the overall employment of bus drivers may remain relatively small. Similar examples can be found in green manufacturing (i.e. 'Cutting, Punching and Press Machine Operators', 'Industrial Machinery Mechanics', 'Shipping, receiving and traffic clerks'). These examples indicate that there is a trade-off between using more inclusive criteria of green occupations and overestimation of potential employment opportunities in green occupations.

Renewable Energy Generation sector has a benefit of having occupations relatively more exclusive to renewable energy sector such as Solar photovoltaic installers, wind turbine technicians, or power plant operators in comparison to others. Dlerdorff et al. defines Renewable Energy Generation sector as being "at the heart of most green discussions."

In addition, we examine the potential transition of fossil fuel workers to different green sectors in the Supplementary Material figure S13, we use a broader definition of green occupations by incorporating different types of green occupations to project the transition rates of fossil fuel workers.

Figure 13: Transition to different types of green jobs

3. The disaggregation is also important because oil and gas extraction workers tend to be more itinerant than coal workers. And both coal plant and refinery workers are less mobile. If that is so, what are the implications?

Authors: We agree with the reviewer that there may be variations in latent mobility among fossil fuel workers based on sub-sectors, and that this could have more specific policy implications. Indeed, among the extraction workers, there are differences in occupational composition that reflect the specificities of each sub-sector. For example, some occupations are more specific to coal mining (e.g., Roof Bolters, Mining (47-5043)), while others are heavily reliant on the oil and gas extraction sector (e.g., Derrick Operators, Oil and Gas (47-5011)).

While we agree with the reviewer's point, it is important to note that, given the current data availability, our regression analysis can only capture the average mobility of fossil fuel workers and does not differentiate the latent mobility of workers by sub-sectors. The J2JOD data from the US Census Bureau, which is used to estimate the regression model (Fig. 1B), provides information on worker mobility solely at the industry level, using 2-digit NAICS codes that encompass both coal mining and oil and gas extraction workers. We model the average mobility of fossil fuel workers using this regression, then apply it to the occupation level data (skill similarity, employment) to predict the proportion of fossil fuel workers transitioning to green jobs. Thus, while we account for some occupational specificities, our analysis does not capture the variation in latent mobility within fossil fuel workers by subsectors. To address this limitation, we have added the following in the discussion section to acknowledge the potential variations in latent mobility among fossil fuel workers by sub-sectors:

First, we have assumed that all of today's fossil fuel workers will want to transition to green jobs. However, there may be additional social barriers including workers' preferences, identity, culture, and economic outlook. Second, we have assumed that fossil fuel workers share the same level of latent mobility. We capture the average mobility of fossil fuel workers using the industry level mobility data (Fig 1B, J2JOD data), then apply this to the occupation level data (skill similarity, employment) to estimate the proportion of workers transitioning to green jobs. However, it is important to note that the latent mobility of workers may vary among fossil fuel workers, particularly across sub-sectors, influenced by socio-economic or cultural factors.

4. Your approach to JT is somewhat narrow and akin to economic development. I would recommend that you explore the literature, particularly that which relates to workers and JT more closely. None of the JT references you use deal with workers. Some deal with energy transition and some explore JT with little or partially informed understanding of JT. Only Pai et al., address workers and JT and, even then, in terms of the polyvalent approach to energy justice.

Authors: This is a fair concern. Indeed, the broader literature on Just Transition is holistic and connects to issues of environmental and climate justice, equity, and so forth. For instance, a recent review from Wang and Lo (2021) lists five conceptualizations of the concept of JT, the labor angle being only one of them.

To address this, we edited the paper in two ways. First, we now embed our study in the broader literature on a Just Transition. We highlight related yet distinct topics, such as environmental justice. Second, we strengthened our review of the literature on the nexus workers-fossil economy-Just Transition. These edits include the addition of the following text:

The Just Transition is an umbrella concept that connects both normative concerns about the ethics of existing and alternative energy systems as well as positive debates about the governance underpinning the clean energy transition (Wang and Lo 2021). Among these, the challenge faced by fossil fuel workers has received increasing attention as one of the constituencies that are directly affected by policy decisions (Cha 2017, Pollin and Callaci 2019, Vanatta et al. 2022, Gazmararian and Tingley 2023).

5. JT proposals (from the very beginning) recognized the centrality of relocation, employment/pension security, schooling and retraining, benefits and a green industrial policy. They realized that the places of sunseting and emerging employment did not always align. The alignment of fossil fuel employment with renewables employment is one solution but not the only and probably not even the most likely solution. People may choose other jobs, e.g., manufacturing or working in the service sector doing technical jobs. For that you need some intentionality in your JT policy. Otherwise it is better to use another term, such as labor market dynamics or economic redeployment.

Authors: We agree that focusing on fossil fuel workers' transition to green industry jobs represents only a subset of the possible ways to absorb fossil fuel workers into other parts of the economy. To address this, we perform a new analysis examining the possibility of fossil fuel workers transitioning to different industries, including the manufacturing industry.

In Figure 4C, we show that fossil-fuel workers' predicted transition to six other sectors: construction, manufacturing, transportation and warehousing, which have the highest skill similarities, and three other industries with the lowest skill similarities with fossil fuel industry. To estimate the fossil fuel workers' transition rates to these industries, we identify the core occupations of each industry using national employment projection data by BLS. Then, using these occupations and the predicted value of their

employment in 2029, along with the regression model from Figure 1B, model (5), we project the potential transition from fossil fuel sector to these industries.

We find that the fossil fuel workers' predicted transition rates to the manufacturing and construction sector is around 3%, which is higher than their transition rate to the renewable energy sector, assuming no major policy interventions (baseline scenario in Figure 4C). For convenience, we provide screenshots of the results here:

We also agree with the reviewer that the co-location of emerging and fading jobs is, in general, not a new topic, but one that is discussed in economics and migration literature more broadly. And yet, the issue of colocation does not appear to have trickled into public discourse around a Just Transition for fossil fuel workers and there remains a need to quantify the barriers that produce friction. For example, a recent ABC News article from May 15, 2023 is entitled “Displaced fossil fuel workers struggling as CA shifts to clean energy, study shows.” As another example, the Natural Resources Defense Council (NRDC) published an article on May 10, 2023, highlighting the potential for fossil fuel workers to transition to green industry jobs based only on the perspective of skills. Both of these recent articles represent public discourse both in the news media and from the perspective of NGOs who are implementing programs on the ground.

REVIEWERS' COMMENTS

Reviewer #1 (Remarks to the Author):

Many thanks for a professional and comprehensive revision! Congratulations on this important work.

Reviewer #3 (Remarks to the Author):

The revised version is more sensitive to the underlying dynamics of location. I am supportive of the publication of the paper and would recommend that the authors make it clearer that they are dealing with one aspect that JT policymakers ought to take into account and, second, that a JT policy is a political decision that requires intentional interventions to address workers and communities as social entities more so than economic entities.

Response to Reviewers:

We thank the reviewers for their constructive feedback regarding our work “**Quantified Barriers to a Just Transition for US Fossil Fuel Workers**” submitted for consideration for publication with *Nature Communications* (NCOMMS-23-07759). In the following, we address each point raised by the reviewers. Reviewers’ comments are presented in black font while authors’ replies are provided in blue font. We specify each change made to the main text and supplementary materials in the revised submission. We were able to address all reviewer comments.

Reviewer #1 (Remarks to the Author):

Many thanks for a professional and comprehensive revision! Congratulations on this important work.

Authors: Thank you for helping to improve our manuscript!

Reviewer #3 (Remarks to the Author):

The revised version is more sensitive to the underlying dynamics of location. I am supportive of the publication of the paper and would recommend that the authors make it clearer that they are dealing with one aspect that JT policymakers ought to take into account and, second, that a JT policy is a political decision that requires intentional interventions to address workers and communities as social entities more so than economic entities.

Authors: We agree. We have revised our Discussion section to include both of these points.

The first paragraph in the Discussion section now reads:

Can a Just Transition be achieved by transitioning fossil fuel workers to new green jobs? The transferability of fossil fuel workers' skills is one factor in their possible transition to other industries. Largely, stakeholders assume that fossil fuel workers have the skills for green jobs while ignoring where green job growth might occur. Our results highlight that fossil fuel workers have greater skill similarity to green industry occupations than to other industries, but further re-skilling might improve the transition. However, the more significant barrier to a Just Transition will be lacking co-location between today's fossil fuel workers and emerging green jobs. Historically, fossil fuel

workers have not exhibited the geospatial mobility necessary to absorb today's fossil fuel workers into the green industry.

We added the following sentence to the limitations paragraph:

Finally, other barriers---particularly social barriers beyond employment---may shape a Just Transition but are not addressed in this study.

And we have revised the concluding paragraph to include:

To be successful, this transition requires a high degree of skill similarity and geographical congruence between green and fossil fuel jobs---in addition to solutions for other non-economic social barriers.